# Joint single-cell multiomic analysis in Wnt3a induced asymmetric stem cell division

Zhongxing Sun[1,3], Yin Tang[1,3], Yanjun Zhang [1,3], Yuan Fang[1,3], Junqi Jia[1], Weiwu Zeng[1] & Dong Fang [1,2✉]

Wnt signaling usually functions through a spatial gradient. Localized Wnt3a signaling can induce the asymmetric division of mouse embryonic stem cells, where proximal daughter cells maintain self-renewal and distal daughter cells acquire hallmarks of differentiation. Here, we develop an approach, same cell epigenome and transcriptome sequencing, to jointly profile the epigenome and transcriptome in the same single cell. Utilizing this method, we profiled H3K27me3 and H3K4me3 levels along with gene expression in mouse embryonic stem cells with localized Wnt3a signaling, revealing the cell type-specific maps of the epigenome and transcriptome in divided daughter cells. H3K27me3, but not H3K4me3, is correlated with gene expression changes during asymmetric cell division. Furthermore, cell clusters identified by H3K27me3 recapitulate the corresponding clusters defined by gene expression. Our study provides a convenient method to jointly profile the epigenome and transcriptome in the same cell and reveals mechanistic insights into the gene regulatory programs that maintain and reset stem cell fate during differentiation.

[1] Zhejiang Provincial Key Laboratory for Cancer Molecular Cell Biology, Life Sciences Institute, Zhejiang University, Hangzhou, Zhejiang 310058, China. [2] Department of Medical Oncology, Key Laboratory of Cancer Prevention and Intervention, Ministry of Education, The Second Affiliated Hospital, Zhejiang University School of Medicine, Hangzhou, Zhejiang 310058, China. [3]These authors contributed equally: Zhongxing Sun, Yin Tang, Yanjun Zhang, Yuan Fang. ✉email: dfang@zju.edu.cn

Stem cells undergo self-renewal to maintain their pluripotency but also give rise to lineage-specific daughter cells that generate differentiated cells[1,2]. Through this asymmetric cell division process, stem cells reside in cellular niches that impart chemical and physical signals to divide asymmetrically, generating one stem cell and one differentiated cell from the parental cell. Stem cells can segregate their cell fate determination components into one of the two daughter cells to accomplish this function. Alternatively, they adopt a positioning strategy in which one daughter cell moves away from the stem cell niche and undergoes differentiation[3–5].

Wnt signaling is conserved from metazoans to vertebrates and plays important roles in stemness maintenance, development, cell survival, tumorigenesis, tumor metastasis, and cellular metabolism[6,7]. Wnt ligands, which comprise a large family of hydrophobic glycoproteins, are usually secreted locally to form an adjacent gradient[8]. Wnt ligands are inactivated if tagged with canonical protein tags, and thus in vivo studies of Wnt signaling are difficult[9]. In addition, our understanding of Wnt signaling is mainly based on the manipulation of the whole population of cells using biochemical or genetic approaches[4]. Mouse embryonic stem cells (mESCs) are cultured in supporting medium that activates Wnt signaling globally to maintain pluripotency[10]. Moreover, inhibition of Wnt signaling leads to the differentiation of mESCs toward an epiblast stem cell-like status[11]. Thus, mESCs represent a powerful model to study the functions of localized Wnt signaling rather than global signaling. Previous studies have shown that Wnt3a-coated beads provide a spatially restricted Wnt signal[12,13]. This localized Wnt signal is transmitted by the Wnt receptor Lrp6 and coordinates with ionotropic glutamate receptor activity, leading to membrane protrusion[14]. After one cell cycle, two daughter cells asymmetrically divide. Cells proximal to Wnt3a beads retain high expression levels of pluripotency markers, such as *Nanog* and *Rex1*. Distal cells exhibit a progressive differentiation status. Parental and newly synthesized histones are segregated into proximal and distal cells with a nonoverlapping pattern, resulting in differential inheritance of histones in two daughter cells[15]. Using single-cell-based high-resolution imaging, several lines of evidence have been proposed to address the response to Wnt signaling. However, the detailed transcriptional and epigenomic changes underlying this asymmetric cell division have not been elucidated.

Recent studies have shown that histone modifications, one of the storage vehicles for epigenetic information, are dramatically changed during early development, germ cell reprogramming, pluripotent cell generation, and stem cell differentiation[16–21]. The profile of the epigenome has mainly been identified using chromatin immunoprecipitation followed by massively parallel sequencing (ChIP-seq)[22]. A large number of cells are required for canonical ChIP-seq, and thus this assay is difficult to conduct with limited input samples[23]. Decreasing the input sample size would unlock a unique aspect of this application. Single-cell level epigenomic and transcriptional profiles can be used to classify lineage-specific regulatory elements, define differentiation trajectories, and identify rare spatiotemporal progenitor cells[24,25]. Over the last decade, several low-input epigenomic profiling methods have been successfully developed, including but not limited to STAR ChIP[18], MOWChIP[26], Drop-ChIP[27], ULI-NChIP[28], Nano-ChIP-seq[29], ChIL-Seq[30], ACT-Seq[31], scChIP-Seq[32], scChIC-seq[33], TAF-ChIP[34], muChIP-seq[35], LIFE-ChIP-seq[36], TCL[37], SurfaceChIP-seq[38], CUT&RUN[39], CUT&Tag[40], CoBATCH[41], and it-ChIP[42]. With the expansion of single-cell technologies in epigenome and transcriptome profiling, emerging needs are to map the epigenome and transcriptome in the same sample. Using this approach, we can save limited samples and, more importantly, avoid batch effects during the experiments. Paired-Tag and

CoTECH were recently established to profile histone modifications and gene expression in single nuclei as a method for this joint analysis[43,44]. These techniques provide the nuclear RNA transcriptome with a comparable number of genes as detected using 10x genomics[43–45].

In this work, we develop SET-seq (same cell epigenome and transcriptome sequencing), which extends the directed tagmentation of mRNA/cDNA hybrids[46,47] and chromatin[40]. SET-seq can be used to simultaneously profile cytoplasmic RNA expression and epigenomic information in a limited number of cells, even in single cells. Using this method, we jointly map the histone modifications and transcriptome of mESCs that undergo Wnt3a-induced asymmetric cell division. Our results reveal cell type-specific maps of the epigenome and transcriptome in asymmetrically divided daughter cells. Additionally, H3K27me3, but not H3K4me3, is rapidly altered and displays a strong correlation with gene expression during asymmetric cell division. Knockout of *Aebp2*, the regulatory element in PRC2 (Polycomb Repressive Complex 2) that is responsible for the methylation of H3K27me3, increases the ratio of daughter cells asymmetrically expressing *Nanog-mCherry*. Together, our results provide mechanistic insights into the gene regulatory programs required for maintaining and resetting stem cell fate during differentiation.

## Results

**The mRNA/cDNA hybrid is directly tagmentated in a reproducible and time-saving manner.** We aimed to separate nuclear DNA and cytoplasmic RNA before library preparation to map the genome-wide transcriptional and epigenetic profiles in the same cell (Fig. 1a). After cell lysis, cytoplasmic RNA and nuclei were separated using concanavalin A (ConA)-coated magnetic beads. The ConA bead-bound nuclei were subjected to antibody binding and tagmentation with CUT&Tag[40], while the cytoplasmic RNA was reverse transcribed and used for direct tagmentation[46,47]. The epigenomic and transcriptional libraries from the same sample were then constructed by performing indexed PCR. We took advantage of the direct tagmentation of Tn5 to reduce the complexity and cost of library construction. The usage of Tn5 for tagmentation of mRNA/cDNA hybrids is recently developed[46,47], although the Tn5 transposome has been widely used to profile double-stranded DNA for years. Therefore, we tested different reaction conditions to improve the reproducibility and stability of transcriptional libraries.

First, we analyzed the effects of temperature and time on the tagmentation and sequencing results. We used mRNA/cDNA hybrids that were reverse transcribed from the total RNA of mESCs by oligo dT as the templates for tagmentation. The mRNA/cDNA hybrids were tagmentated to approximately 100 to 400 base pairs when the reactions were incubated at 4 °C, 16 °C, and 37 °C for 5 to 30 min (Supplementary Fig. S1a). The tagemented fragments became shorter with higher temperatures and prolonged incubation times. We then constructed and sequenced libraries produced by tagmentation at 16 °C and 37 °C for 5 to 30 min because, when samples were tagmentated at these temperatures, less time was needed for library preparation compared to tagmentation at 4 °C. Two biologically independent libraries were constructed and sequenced for each condition to increase the sequencing strength. These libraries showed strong correlations with conventional ligation-based RNA-seq library (Methods) and among each library (Fig. 1b). We also detected a similar number of genes (Supplementary Fig. S1b and c) and similar gene expression profiles (Supplementary Fig. S1d). Moreover, we found that a prolonged tagmentation time at 37 °C slightly reduced the ratio of detected reads at exons compared to those at introns (Supplementary Fig. S1e). Then, we chose 37 °C

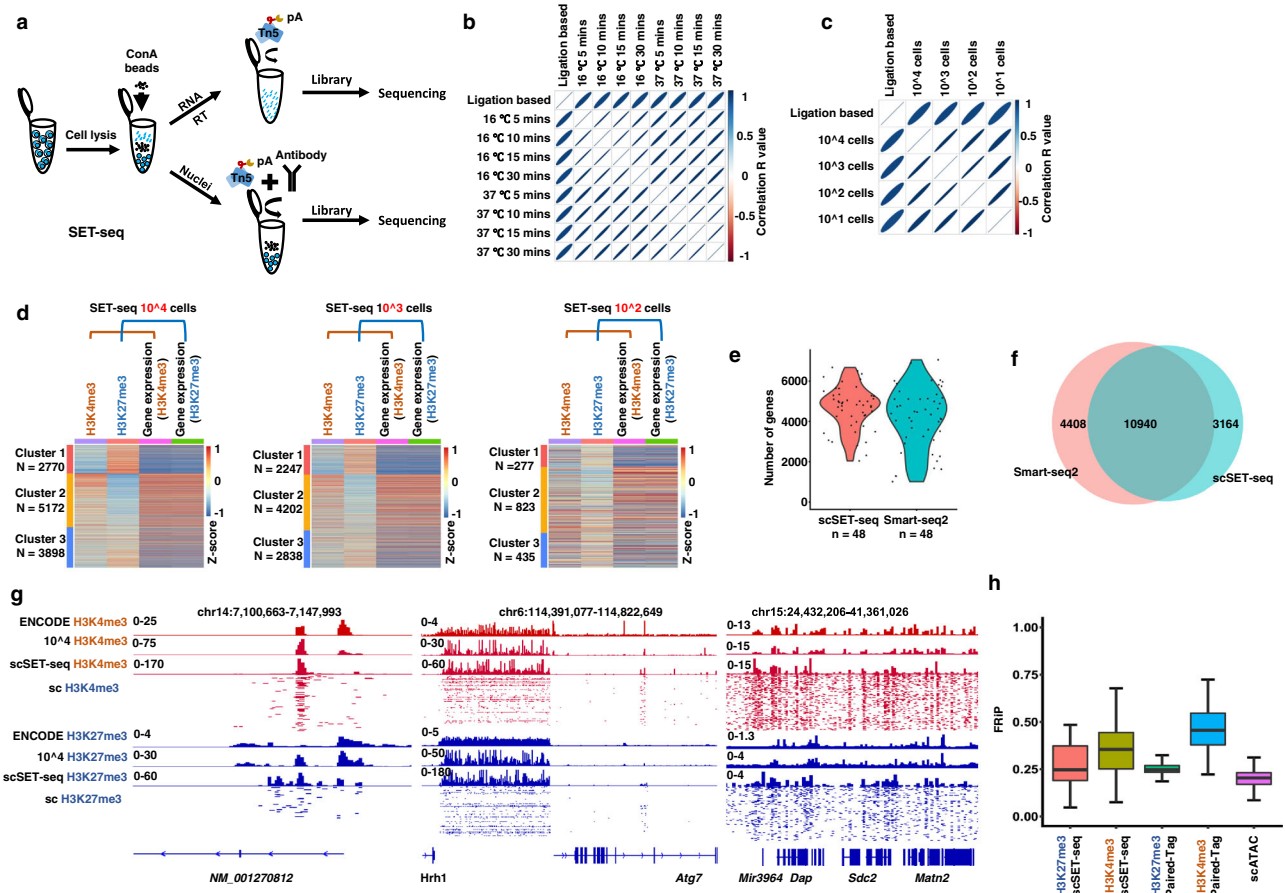

**Fig. 1 SET-seq can be used to profile epigenome and transcriptome in the same sample. a** Schema showing the process of SET-seq. **b** The Pearson correlations among gene expression libraries constructed from different tagmentation time and temperatures. **c** The Pearson correlations among gene expression libraries constructed from different numbers of cells. **d** Heatmaps showing the clustering results of H3K27me3 and H3K4me3 SET-seq. Genes, at which the epigenomic and transcriptional signals were detected, were clustered by hierarchical clustering algorithms using row scaled signal scores. Heatmaps were plotted with column scaled signal scores. **e** Number of detected genes in scSET-seq and Smart-seq2 in 48 mESCs. Smart-seq2 data were downloaded from GSE151334 [https://www.encodeproject.org/experiments/ENCSR059MBO/] and 48 cells were randomly selected. **f** Venn diagram showing the overall detected genes between scSET-seq and Smart-seq2 in 48 mESCs. Smart-seq2 data were as in **e**. **g** Examples of IGV views of H3K27me3 and H3K4me3 scSET-seq. Signals of histone marks from 200 individual single cells were shown below the signals from aggregated 480 cells. H3K4me3 and H3K27me3 SET-seq results from 10,000 cells (10^4 H3K4me3 and 10^4 H3K27me3) were shown as controls. **h** The fraction of reads in peaks (FRiPs) of scATAC-seq and scSET-seq. scATAC-seq data were downloaded from GSE100033 [https://www.ncbi.nlm.nih.gov/geo/query/acc.cgi?acc=GSE151334]. scSET-seq cells were filtered with a mapping ratio >10%. Paired-Tag and 10x genomics data was same as previous reported[43]. The boxes were drawn from lower quartile (Q1) to upper quartile (Q3) with the middle line denoting the median, and whiskers with maximum 1.5 IQR (interquartile range). $n = 45$ (H3K27me3 scSET-seq), 200 (H3K4me3 scSET-seq), 447 (H3K27me3 Paired-Tag), 1,659 (H3K4me3 Paired-Tag), and 93 (scATAC-seq) cells. Source data are provided as a Source Data file.

and 5 min as the tagmentation conditions for subsequent experiments to reduce the reaction time and avoid the loss of exons.

Second, we evaluated the potential effect of the amount of Tn5 (25, 50, or 100 ng/μl) on the sequencing results. Higher amounts of Tn5 resulted in slightly shorter fragment lengths after tagmentation (Supplementary Fig. S2a). The correlations among sequencing results from different amounts of Tn5 were quite high (Supplementary Fig. S2b). In addition, the identified gene numbers, gene expression profiles, and ratios of reads mapped to exons and introns were similar when different amounts of Tn5 were used (Supplementary Fig. S2c–e), suggesting that the amount of Tn5, ranging from 25 to 100 ng/μl, had a minor effect on the sequencing results.

Third, using a series of 100-fold dilutions starting from 300 ng of total RNA, we analyzed the minimum amount of RNA needed for sequencing library construction. We obtained sufficient DNA for sequencing with as low as 30 pg of total RNA, which was

similar to the amount of total RNA in a single mESC[48] (Supplementary Fig. S3a). We sequenced the libraries constructed from 300 ng, 3 ng, and 30 pg of total RNA. The correlations among each library were high, and the ratios of reads mapped to exons and introns were stable (Supplementary Fig. S3b and c). A total of 17,097, 14,659, and 9,371 genes were detected from 300 ng, 3 ng, and 30 pg of total RNA, respectively (Supplementary Fig. S3d), indicating that fewer genes were identified with less starting RNA. Together, we established a timely and reproducible reaction condition, incubation at 37 °C for 5 min, for the direct tagmentation of mRNA/cDNA hybrids, and this condition might be applied at the single-cell level.

**SET-seq can be used to jointly profile the epigenome and transcriptome in a low number of cells, even in single cells.** Thereafter, we applied this reaction condition to profile cytoplasmic RNA extracted using the SET-seq method in vivo. The sequencing results were stable and highly correlated when

10,000 1,000, 100 and 10 cells were used as starting materials (Fig. 1c and Supplementary Fig. S4a–c). Additionally, we analyzed the stabilities of epigenomic profiles of SET-seq using H3K27me3 antibodies in 10–10,000 cells and performed SET-seq to analyze H3K27me3 profiles in four replicates. The peak distributions of H3K27me3 SET-seq were similar to H3K27me3 ChIP-seq in the ENCODE project ENCSR059MBO [https://www.encodeproject.org/experiments/ENCSR059MBO/] (Input control data ENCSR326ULS [https://www.encodeproject.org/experiments/ENCSR326ULS/ENCSR326ULS) (Supplementary Fig. S5a). Moreover, the SET-seq signals from 10, 100, and 1,000 cells were highly enriched at the peaks corresponding to 10,000 cells (Supplementary Fig. S5b). We measured the SET-seq accuracies by constructing a receiver operating characteristic curve (ROC)[49] and detected high area under the curve (AUC) values ranging from 0.801 to 0.982 (Supplementary Fig. S5c). These results indicate that the transcriptional and epigenomic libraries constructed using SET-seq are reproducible when different numbers of cells are used as starting material.

We applied H3K4me3 and H3K27me3 SET-seq in 100 to 10,000 cells. We assigned epigenomic signals to the closest promoter and then selected genes with both epigenomic and transcriptional signals in H3K27me3 and H3K4me3 SET-seq to jointly compare gene expression levels and enrichments of histone marks. We then scaled the epigenomic and transcriptional signals in the same sample with the $Z$ score and clustered the genes based on H3K4me3 and H3K27me3 signals. The corresponding expression levels of each gene were marked accordingly (Fig. 1d). Three major clusters were generated from H3K4me3 and H3K27me3 signals. In Cluster 1, genes showing low H3K4me3 and high H3K27me3 enrichment were expressed at low levels. In Cluster 2, high H3K4me3 and low H3K27me3 enrichment were detected at these genes, which were expressed at high levels. Moderate H3K4me3 and H3K27me3 levels were detected as bivalent histone marks in Cluster 3 genes, which were expressed at low levels. This pattern recapitulated the correlations between histone marks and gene expression, where the active histone mark H3K4me3 correlated with high levels of gene expression, the repressive histone mark H3K27me3 correlated with low gene expression levels, and bivalent H3K4me3 and H3K27me3 were associated with repressed genes in mESCs. In addition, we performed a Gene Ontology (GO) analysis using the clustered genes from 10,000 cells (Supplementary Fig. S5d). Genes in Cluster 1 with high H3K27me3 levels were mainly enriched in ion transport-associated terms. Genes in Cluster 2 with high H3K4me3 levels were enriched in cell cycle- and signal transduction-linked terms. Cluster 3 genes with both H3K27me3 and H3K4me3 signals were annotated as differentiation- and development-associated terms. This result was consistent with the findings that bivalent genes, which were enriched with both H3K27me3 and H3K4me3, were mainly differentiation marker genes in mESCs[50]. Because two histone marks, H3K4me3 and H3K27me3, along with gene expression needed to be detected for the same gene, fewer genes were detected when 100 cells were used as the starting material. The classified clusters from 100 cells were less distinct than the clusters obtained from larger numbers of cells. We detected approximately 2,000 to 5,000 genes in one cluster when 1000 or 10,000 cells were used. When 100 cells were used as the starting material, the genes with detected H3K27me3, H3K4me3 and gene expression signals decreased to approximately 300 to 800 in each cluster. These data indicate that when bulk cells are used to detect both H3K4me3 and H3K27me3 along with gene expression signals, at least 1,000 cells are optimal for this specific application.

We profiled gene expression in 48 single mESCs using single-cell SET-seq (scSET-seq) in 3 independent experiments to extend our SET-seq method to an even lower number of cells. Tn5

transposomes with different combinations of i5 and i7 barcodes were applied to facilitate library preparation. Moreover, the structure of the DNA library was designed to be similar to TruSeq libraries to enable it to be sequenced with other conventional Illumina libraries together (Methods)[51]. Compared to 4,206 genes detected using Smart-seq2 GSE151334 [https://www.ncbi.nlm.nih.gov/geo/query/acc.cgi?acc=GSE151334], which revealed the most genes per cell among commercially available approaches, scSET-seq had a similar sensitivity, capturing an average of 4,665 genes per cell (Fig. 1e). The diversity of the total detected genes across 48 single cells was similar between scSET-seq and Smart-seq2 (Fig. 1f). These two methods reached a plateau of saturation at approximately 2 million total reads (Supplementary Fig. S6a). In addition, we compared the fractions of intragenic and intronic reads among scSET-seq, Smart-seq2, Paired-Tag[43], and 10x genomics[43]. The library construction methods were similar between scSET-seq and Smart-seq2, which were based on whole cDNA fragments, and between 10x genomics and Paired-Tag, which mainly detected the 3' end of cDNAs. As previously reported[52], the ratios of intragenic reads and intronic reads were both lower in Smart-seq2 data than in 10x genomics data (Supplementary Fig. S6b and c). The scSET-seq and Paired-Tag approaches were similar to Smart-seq2 and 10x genomics based on the ratios of intragenic and intronic reads, respectively. Similar to Smart-seq2, the gene expression libraries of scSET-seq were constructed without unique molecular identifiers (UMIs), which are widely used in 3' RNA–seq techniques, such as 10x genomics, to distinguish unexpected PCR duplications and rare mutation variants.

Furthermore, we applied scSET-seq to H3K27me3 and H3K4me3 in 480 mESCs. Five independent experiments with 96 cells in each batch were performed for H3K27me3 and H3K4me3 scSET-seq (Supplementary Dataset 1). The aggregated single-cell profiles recapitulated the bulk ChIP-seq data, as shown in Integrative Genomics Viewer (IGV) examples (Fig. 1g). We also compared scSET-seq to scATAC-seq and Paired-Tag to determine the accuracy of signals detected in single cells. The fraction of reads in peaks (FRiPs) was comparable and similar between scSET-seq and Paired-Tag, both of which were higher than scATAC-seq (Fig. 1h). We also noticed higher FRiPs for H3K4me3 than for H3K27me3 in both Paired-Tag and scSET-seq. The mean and median numbers of unique fragments detected using H3K27me3 and H3K4me3 scSET-seq (Mean: H3K27me3 2,682, H3K4me3 7,193, Median: H3K27me3 1,633, H3K4me3 4,601) were similar to those detected using Paired-Tag (Mean: H3K27me3 3,992, H3K4me3 7,969, Median: H3K27me3 3,067, H3K4me3 6,225), respectively (Supplementary Fig. S6d). We sequenced more than 2 million reads per cell to reach the saturation of unique fragments recovered. Moreover, we called peaks from the bulk SET-seq data of 10,000 cells, merged scSET-seq results, and ENCODE data for H3K27me3 ENCSR059MBO [https://www.encodeproject.org/experiments/ENCSR059MBO/] and H3K4me3 ENCSR000CGO [https://www.encodeproject.org/experiments/ENCSR000CGO/], respectively. The peaks called from bulk SET-seq largely overlapped with the corresponding ENCODE data. Peaks called from merged scSET-seq data overlapped well with bulk SET-seq and ENCODE data (Supplementary Fig. S6e and f). Together, these results indicate that SET-seq is useful to analyze both the epigenome and transcriptome in the same cells, even in single cells.

**Transcriptome analysis of H3K27me3 and H3K4me3 scSET-seq identifies cell clusters in localized Wnt3a-induced asymmetric mESCs division.** Previous studies have shown that Wnt3a-coated beads induce the asymmetric division of mESCs[12].

Using an *mCherry*-based reporter for *Nanog*, we also detected that when parental cells were attached to Wnt3a-coated beads, the divided daughter cells proximal to beads showed a higher level of *Nanog* than the cells distal to beads (Supplementary Fig. S7a). We further tested the activation mediated by Wnt3a-coated beads by analyzing the expression of pluripotency markers in bulk cells. We seeded cells with a large amount of Wnt3a-coated beads such that almost all cells were attached to Wnt3a beads when we collected samples. Cells treated with the WNT3a protein were used as the positive control. Consistent with previous reports[12], the expression of *Nanog* and *Axin2*, but not *Oct4*, in cells treated with Wnt3a beads was increased compared to cells treated without beads (Supplementary Fig. S7b). We then profiled H3K27me3 and H3K4me3 levels in bulk cells with or without WNT3a treatment. Interestingly, H3K27me3 peaks changed dramatically upon WNT3a treatment, whereas H3K4me3 peaks largely overlapped between untreated and treated cells (Supplementary Fig. S7c). This difference in the change of H3K27me3 and H3K4me3 levels prompted us to further analyze how the epigenome and transcriptome were altered during asymmetric cell division.

Stem cells undergoing asymmetric cell division pose challenges to single-cell analyses because, unlike the majority of cells, a small portion of cells divide symmetrically[12]. The single-cell epigenome and transcriptome provided a unique perspective to illustrate the Waddingtonian landscape of cell fate decisions. We then conducted H3K27me3 and H3K4me3 scSET-seq analyses of mESCs with localized Wnt3a signaling. Since the cells proximal and distal to beads were unable to be sorted using conventional marker-based approaches, such as fluorescence-activated cell sorting (FACS), we manually examined cells adjacent to beads and transferred the proximal and distal daughter cells into 96-well plates to perform indexed scSET-seq. We picked the cells without predetermining the *Nanog* signal to avoid the biased selection of mESCs. Tn5 transposomes with indexed i5 and i7 primers were used to barcode the individual cells (Supplementary Dataset 2). After the direct tagmentation of mRNA/cDNA hybrids and antibody-guide tagmentation of genomic DNA, samples were pooled, PCR indexed, and sequenced (Fig. 2a). We performed H3K27me3 and H3K4me3 scSET-seq in 384 and 368 cells, respectively, generating 1,504 genome-wide profiles. H3K27me3 and H3K4me3 scSET-seq were conducted in 10 and 6 independent experiments, respectively.

We then filtered the cells with gene expression profiles and obtained 335 and 210 cells in H3K27me3 and H3K4me3 scSET-seq that passed the quality control criteria (Methods), respectively (Supplementary Dataset 3). We merged the transcriptional profiles of H3K27me3 and H3K4me3 scSET-seq to improve the reproducibility of dimensional reduction. We performed batch correction by conducting a canonical correlation analysis (CCA) to address technical variances, such as processing and individual variation of single cells. Three main clusters were then identified by the shared nearest neighbor modularity optimization algorithm in Seurat[53] (Fig. 2b). Because we indexed the proximal and distal cells with distinct barcodes, we then marked the clustered cells with proximal and distal tags (Fig. 2c). Interestingly, cells in Cluster 0 were enriched with both proximal and distal cells. Cluster 1 was mainly composed of cells proximal to Wnt3a beads, while cells in Cluster 2 were labeled as distal to Wnt3a beads. We then marked Cluster 1 with the most proximal cells as Proxi, Cluster 2 mainly with the distal cells as Dista, and Cluster 0 with mixed cells as Mix. The GO enrichment analysis showed that marker genes in the Mix cluster were enriched for neuron interaction- and differentiation-associated terms, while marker genes in the Dista and Proxi clusters were enriched for RNA processing (Supplementary Fig. S8a–c). In particular, the GO

terms in the Dista cluster were annotated to several nucleoside triphosphate metabolic process terms. This process participates in erythroid differentiation[54], T cell lineage differentiation[55], and neurogenesis[56]. Differentiation-associated terms may have been overridden by the nucleoside triphosphate metabolic process terms. Previous studies[12] and our bulk cell analysis using RT–PCR (Supplementary Fig. S7b) have shown that *Nanog* and *Rex1* are expressed at high levels in proximal cells. We then analyzed the expression of *Nanog* and *Rex1* among different clusters of cells and found that they were expressed at higher levels in Proxi cells, which were enriched with proximal cells (Supplementary Fig. S9).

We then split the cells based on their profiled histone marks: H3K27me3 (Fig. 2d) and H3K4me3 (Fig. 2e). Not surprisingly, three clusters were segregated well into proximal and distal cells when divided based on H3K27me3 scSET-seq (Fig. 2f) and H3K4me3 scSET-seq (Fig. 2g) data. Furthermore, the expression profiles of marker genes in each cluster were similar between H3K27me3 and H3K4me3 scSET-seq results (Supplementary Fig. S10a–d). Because the two daughter cells were picked as pairs, we projected the cells in the same pair to examine how the daughter cells were distributed among cell clusters (Fig. 2h, i). Most daughter cells were paired between the Proxi and Dista clusters, as well as inside the Mix cluster. No cells were paired inside the Proxi or Dista clusters. Nevertheless, we found that half of the cells in the Proxi/Dista cluster were paired with cells in the Mix cluster, suggesting that one of the daughter cells is at a modest differentiation stage in these pairs. The proportions of clustered cells were 24.4% for the Proxi cluster, 22.2% for the Dista cluster, and 53.4% for the Mix cluster. Notably, we observed a small number of proximal cells in the Dista cluster and distal cells in the Proxi cluster, accounting for approximately 10% of the total cells. These data suggest that these cells are segregated as a reversed asymmetric division. Indeed, previous studies[12,15] showed that approximately 15% of cells asymmetrically underwent reversed cell division, whereas distal cells showed higher *Nanog* signals than proximal cells. Moreover, the ratio of paired cells within the Mix cluster was 56.7% of cells in the Mix cluster. Therefore, the ratio of paired cells in the Mix clusters was 30.3% of the total cells. This result was similar to previous reports[12,15,57] showing that approximately 25–30% of cells divided symmetrically. Other cells comprised approximately 60% of the total cells, similar to the previously reported ratio of asymmetrically divided cells (~60%). The annotated clusters reflected the previously reported cell division modes at the level of ratios of cell behaviors.

We generated pseudotemporal ordering by ranking single cells with marker genes to identify trajectories of cell stages underlying Wnt3a-induced asymmetric cell division. Consistent with the observed cell division pattern, pseudotime reconstruction ordered sequentially Proxi and Dista clusters, with the Mix cluster branching in the middle stage (Fig. 2j). The Proxi cluster was located at the beginning of the trajectory, and the Dista cluster was located at the end of the trajectory. In addition, the Mix cluster, which initiated in the middle of the differentiation stage, was detected in one branch alongside the trajectory. This trajectory indicated differentiation progress from the early Proxi cluster to the late Dista cluster, whereas the Mix cluster occurred as a distinct differentiation lineage that differed from the asymmetrically divided cells. We also labeled cells with their positions relative to Wnt3a beads in the trajectories. Compared with distal cells, proximal cells were enriched at earlier stages. Proximal and distal cells were sorted at the middle branch corresponding to the Mix cluster. These patterns were revealed in both H3K27me3 and H3K4me3 scSET-seq data, further supporting the hypothesis that the sequencing results were reproducible

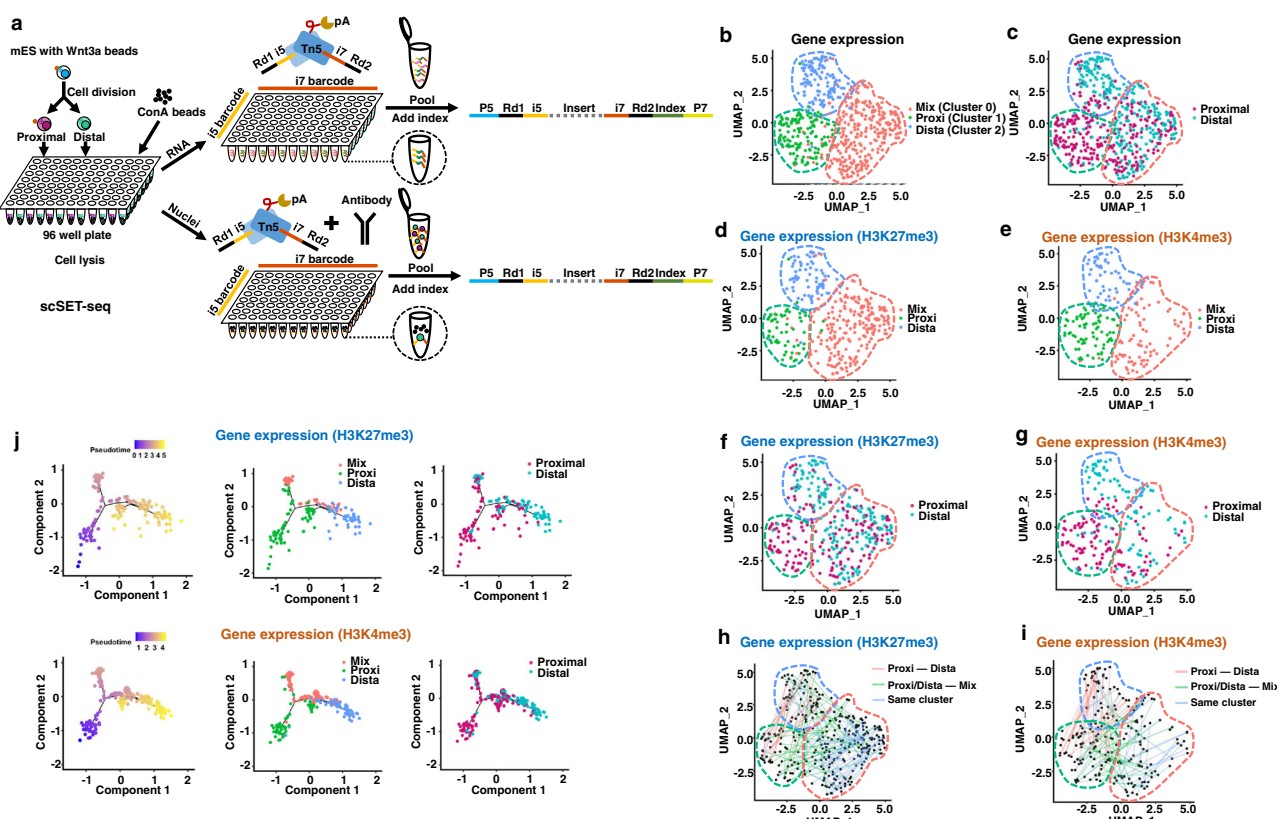

**Fig. 2 scSET-seq is able to identify the cell clusters in Wnt3a beads induced asymmetric cell division in mESCs. a** Schema showing the process of scSET-seq. Epigenome and transcriptome were indexed by transposomes before pooling and library construction. **b** Uniform Manifold Approximation and Projection (UMAP) embedding showing the clustering of cells from H3K27me3 and H3K4me3 scSET-seq by gene expressions. To increase the reproducibility, the gene expressions of H3K27me3 and H3K4me3 scSET-seq were merged for clustering. Each dot represented a single cell. See main text for the definition of clustering name. **c** Colored map showing the relative position of cells to Wnt3a beads. Cells proximal to beads were marked garnet and distal to cells were marked cyan. **d**, **e** UMAP clustered cells were subdivided into H3K27me3 scSET-seq (**d**) and H3K4me3 scSET-seq (**e**). **f**, **g** Colored map showing the relative position of cells to Wnt3a beads in H3K27me3 scSET-seq (**f**) and H3K4me3 scSET-seq (**g**). Cells proximal to beads were marked garnet and distal to cells were marked cyan. **h**, **i** Paired daughter cells were lined up showing their distributions among cell clusters. Two daughter cells from the same parental cell were identified by their index in H3K27me3 scSET-seq (**h**) and H3K4me3 scSET-seq (**i**). Red lines indicated the paired cells were between Proxi and Dista cluster, green lines indicated the two daughter cells were between Proxi/Dista cluster and Mix cluster, and blue lines connected cells inside the same cluster. **j** Pseudotime analysis of single-cell lineages using H3K27me3 and H3K4me3 scSET-seq transcriptional data. The left panels showed the pseudotime scores across the branches. The cell clusters were marked in the middle panels. Relative positions to Wnt3a beads were marked in the right panels. Source data are provided as a Source Data file.

and stable. Moreover, we performed STREAM (single-cell trajectory reconstruction, exploration and mapping) analysis[58] to assign single cells for the reconstruction of developmental trajectories. This analysis recapitulated the branched Mix cluster and early Proxi cluster to the late Dista cluster, further supporting the accuracy of pseudotime reconstruction (Supplementary Fig. S11a). We merged the gene expression data from H3K27me3 and H3K4me3 scSET-seq and then projected several marker genes (*Cracr2a*, *4930438A08Rik*, *Aebp2*, *Rtl1*, *Rpl10a*, *Rnf7*, *Ndufb10*, *Rpl37*, and *Rps4x*) alongside the corresponding pseudotemporal trajectories to precisely compare changes in gene expression (Supplementary Fig. S11b). The marker genes of the Proxi and Dista clusters were expressed at low levels during the branched stages of trajectories, which corresponded to the Mix cluster. Meanwhile, the marker genes of the Mix cluster were expressed at low levels in the trajectory stages corresponding to the Proxi and Dista clusters. Please also note that the clusters in Supplementary Fig. S11b are presented in a one-dimensional plot to show the changes in genes among these three clusters. The Mix cluster, which branched from the Proxi cluster, was not an intermediate between the Proxi and Dista clusters. Together,

using scSET-seq, we were able to identify the cell clusters and sets of marker genes during mESC division with localized Wnt signaling.

**Gene expression is correlated with H3K27me3 levels during mESC division induced by Wnt3a beads.** We aggregated H3K27me3 and H3K4me3 signals in different clusters of cells and visualized their enrichments at the marker genes in each cluster to investigate how histone modifications were detected using scSET-seq. As shown in the IGV views, H3K27me3 and H3K4me3 were distributed around the marker genes and were enriched to different degrees among cell clusters (Fig. 3a). We then sought to annotate coassayed cells based on their epigenomic profiles at the single-cell level. We used the aggregated histone mark signals from each scSET-seq sample, called broad peaks for H3K27me3 and narrow peaks for H3K4me3, and then assigned the epigenomic signals to each gene utilizing the probabilistic topic modeling method in cisTopic[59]. We coembedded epigenomic and transcriptional datasets in the same cell for further analysis (Fig. 3b).

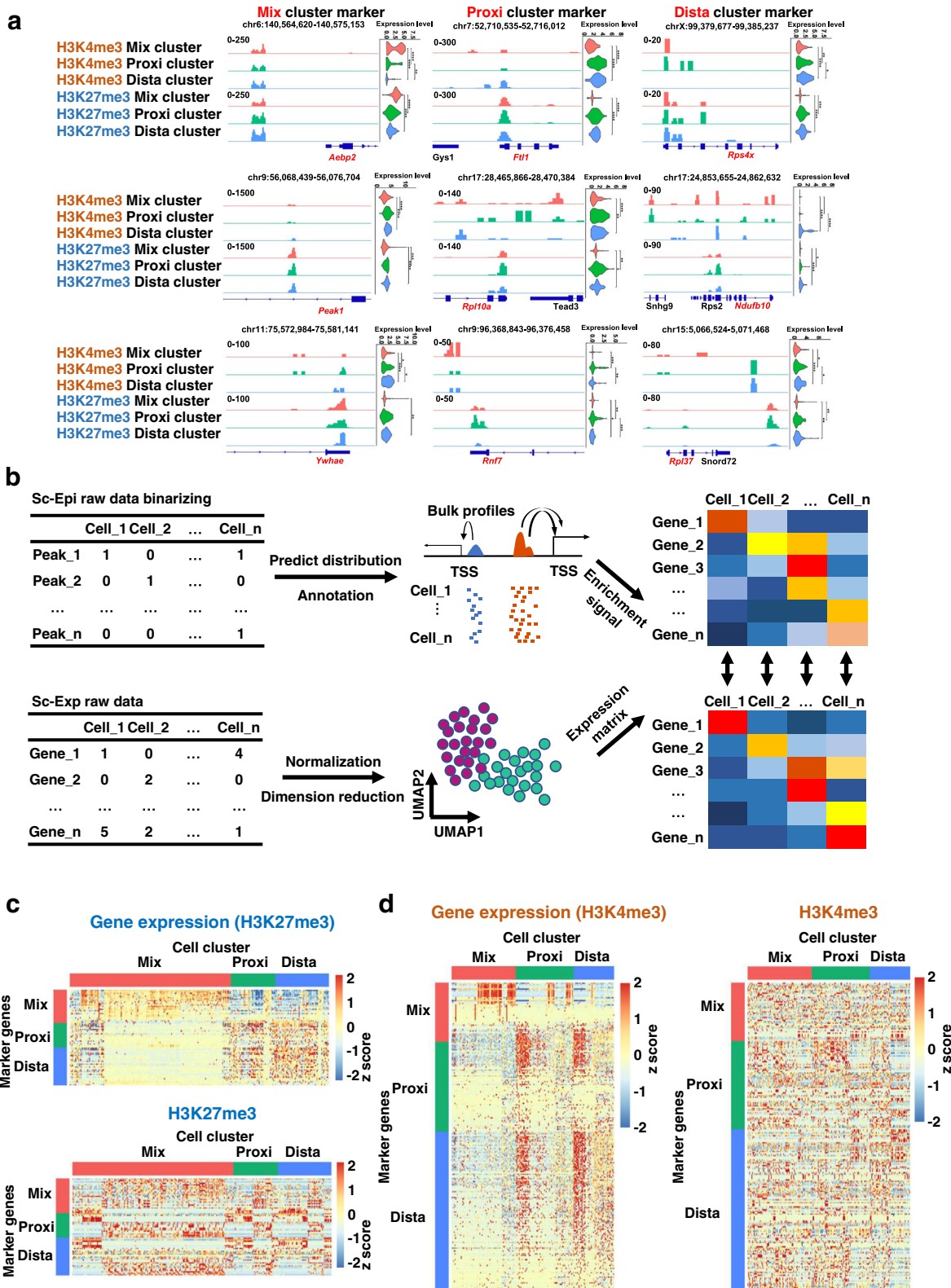

We selected marker genes with detected signals for both the epigenome and transcriptome to visualize the H3K27me3 and H3K4me3 scSET-seq results in heatmaps (Fig. 3c, d). We noticed that fewer marker genes were selected in H3K27me3 scSET-seq data than in H3K4me3 scSET-seq data because H3K27me3 was not detected around the marker genes. H3K27me3 may represent a repressive mark for gene

expression, leading to its low enrichment at marker genes with high expression levels. Gene expression heatmaps showed that cells were clustered well by the marker genes in H3K27me3 and H3K4me3 scSET-seq data. Cells were clustered well when we used H3K27me3 signals at the marker genes to generate the heatmap. However, H3K4me3 was distributed at marker genes in a disordered manner throughout clusters, indicating that

**Fig. 3 Epigenomic profiles of H3K27me3 and H3K4me3 scSET-seq are differently correlated with corresponding gene expressions. a** Joint views showing the IGV examples of aggregated epigenomic signals and violin plots of expression levels. Three marker genes, which were labeled red at the bottom, of each cluster were shown. The epigenomic signals were aggregated from cells of the indicated cluster. *P* values were calculated by Student's t-test, two-sided. $n = 83$ (H3K4me3 Mix cluster), 75 (H3K4me3 Proxi cluster), 52 (H3K4me3 Dista cluster), 208 (H3K27me3 Mix cluster), 58 (H3K27me3 Proxi cluster), and 60 (H3K27me3 Dista cluster) cells *$P < 0.05$, **$P < 0.01$, ***$P < 0.001$, ****$P < 0.0001$. Exact *P* values were provided in Supplementary Dataset 4. **b** Schema showing the process of co-embedding of epigenomic and transcriptional profiles from scSET-seq. See main text for description. **c** Heatmaps showing the co-embedded signals in single cells. Marker genes with both H3K27me3 and transcriptional signals were shown. To assign the H3K27me3 signals, peaks that were called from the aggregated data were preferably annotated to genes within 3 Kb from their transcription starting sites, then to the closest genes within 500 Kb. **d** Same as in **c**, except H3K4me3 scSET-seq data were plotted. Source data are provided as a Source Data file.

H3K4me3 was not altered accordingly with gene expression changes.

Because the signals of the single-cell epigenome were low, preventing us from easily and directly comparing the correlations between epigenetic and transcriptional profiles, we sought to conduct module assignments calculated based on the signals detected in the transcriptome and epigenome. We integrated genes into functional modules by performing a weighted gene co-expression network analysis (WGCNA) and then calculated the correlations between epigenetic and transcriptional profiles (Supplementary Dataset 5). Similar to the heatmap visualizations of gene expression and histone marks (Fig. 3c, d), H3K27me3-signals were sorted into fewer modules than H3K4me3 signals (Supplementary Fig. S12a–d). H3K27me3 modules were correlated with the corresponding gene expression modules, showing remarkable conservation of H3K27me3 and gene expression (Fig. 4a and Supplementary Dataset 5). H3K4me3 modules were less correlated with corresponding gene expression modules (Fig. 4b and Supplementary Dataset 5). In addition, the ratio of significantly correlated modules was higher in H3K27me3 scSET-seq data (21.5%) than H3K4me3 scSET-seq data (10.7%).

We calculated the total signals for cluster marker genes in each cell to further analyze how H3K27me3 and H3K4me3 were distributed in individual single cells. The H3K27me3 signal was high in the Mix cluster of cells when marker genes of three clusters were calculated (Fig. 4c and d). Notably, H3K27me3 signals at Mix cluster marker genes were higher in cells of the Proxi cluster than in cells of the Dista cluster. Unlike H3K27me3, H3K4me3 signals at marker genes were not significantly changed among the three clusters of cells (Fig. 4e, f). Furthermore, we aggregated the epigenomic profiles for each cluster of cells to detect the changes in the epigenome at the total level. H3K27me3 was enriched at the highest level at marker genes of the Mix cluster, while H3K4me3 was similarly enriched across each cluster of marker genes (Supplementary Fig. S13a). Moreover, we marked the cells through pseudotime trajectories with H3K27me3 and H3K4me3 signals of all marker genes. High H3K27me3 signals were detected at the branched stage corresponding to the Mix cluster compared with the start and end points, which corresponded to the Proxi and Dista clusters, respectively (Supplementary Fig. S13b). Meanwhile, H3K4me3 signals were steadily distributed across the trajectories (Supplementary Fig. S13c). Taken together, these results suggest that, unlike H3K4me3, H3K27me3 is correlated with changes in gene expression during Wnt3a-induced asymmetric cell division, playing important roles in the different behaviors of daughter cells.

**Clusters defined by H3K27me3 recapitulate the cell clusters classified by gene expression.** We performed cluster annotations using the epigenomic datasets from H3K27me3 and H3K4me3 scSET-seq to further investigate the epigenomic changes during asymmetric cell division. After dimensional reduction, cells were classified into three main clusters based on H3K27me3 signals (Fig. 5a). Cluster 0 was composed of both proximal and distal

cells, Cluster 1 was primarily composed of proximal cells, and Cluster 2 was mainly composed of distal cells (Fig. 5b). Thereafter, Clusters 0, 1, and 2 were named Mix, Proxi, and Dista, respectively. Similar to the clusters defined by the corresponding gene expression obtained from H3K27me3 scSET-seq data, two divided daughter cells were distributed mainly between Proxi and Dista clusters, between Proxi/Dista and Mix clusters, and inside Mix clusters (Fig. 5c). Accordingly, dimensional reduction by H3K4me3 signals clustered cells based on their positions relative to Wnt3a beads (Fig. 5d, e). However, two daughter cells were distributed completely between Proxi and Dista clusters and inside the Mix cluster (Fig. 5f), which was different from the results obtained from H3K27me3 scSET-seq.

We projected cluster annotations across the cell identities between epigenetic and transcriptional clusters to explore how our predictions of cellular classification from the epigenome and transcriptome were integrated. Cellular clusters predicted from H3K27me3 levels were mainly projected to their corresponding clusters identified by gene expression (Fig. 5g). Unlike H3K27me3, cells clustered based on H3K4me3 levels were mainly projected to the Mix cluster identified by the transcriptome analysis (Fig. 5h). We then calculated the projected ratios of each cluster to more conclusively compare the enrichment (Fig. 5I, j). We used the percentage of cells classified by corresponding gene expression as the random distribution control. If cells in a cluster defined by the epigenomic signal were randomly distributed, they would recapitulate the ratio of clusters identified by gene expression. Otherwise, cells would be enriched in this specific cluster, which was different from the random distribution. Ratios of clustered cells obtained from H3K27me3 data, but not H3K4me3 data, were significantly different from the random distribution control. Moreover, cells classified by H3K27me3 levels were mainly enriched in the corresponding clusters identified by gene expression in the H3K27me3 scSET-seq analysis. On the other hand, cells in each cluster classified by H3K4me3 levels were mostly enriched in the Mix cluster identified based on gene expression, similar to the random distribution. Together, these joint analyses of scSET-seq datasets underscored the conserved representation of the major cell types across histone modifications and gene expression, further suggesting that H3K27me3 correlated with gene expression during Wnt3a-induced asymmetric cell division.

**Knockout of *Aebp2* increases the ratio of daughter cells asymmetrically expressing *Nanog*.** Because we observed that H3K27me3 correlated with changes in gene expression and cell clusters during asymmetric cell division, we carefully examined the expression levels of components of PRC2 that were responsible for the methylation of H3K27me3. PRC2 is composed of core proteins, including EED, EZH2, SUZ12, RBBP4, and RBBP7, and regulatory proteins, such as AEBP2 and JARID2. Among the genes encoding proteins in PRC2, only *Aebp2*, one of the Mix cluster marker genes, was expressed at high levels in the Mix

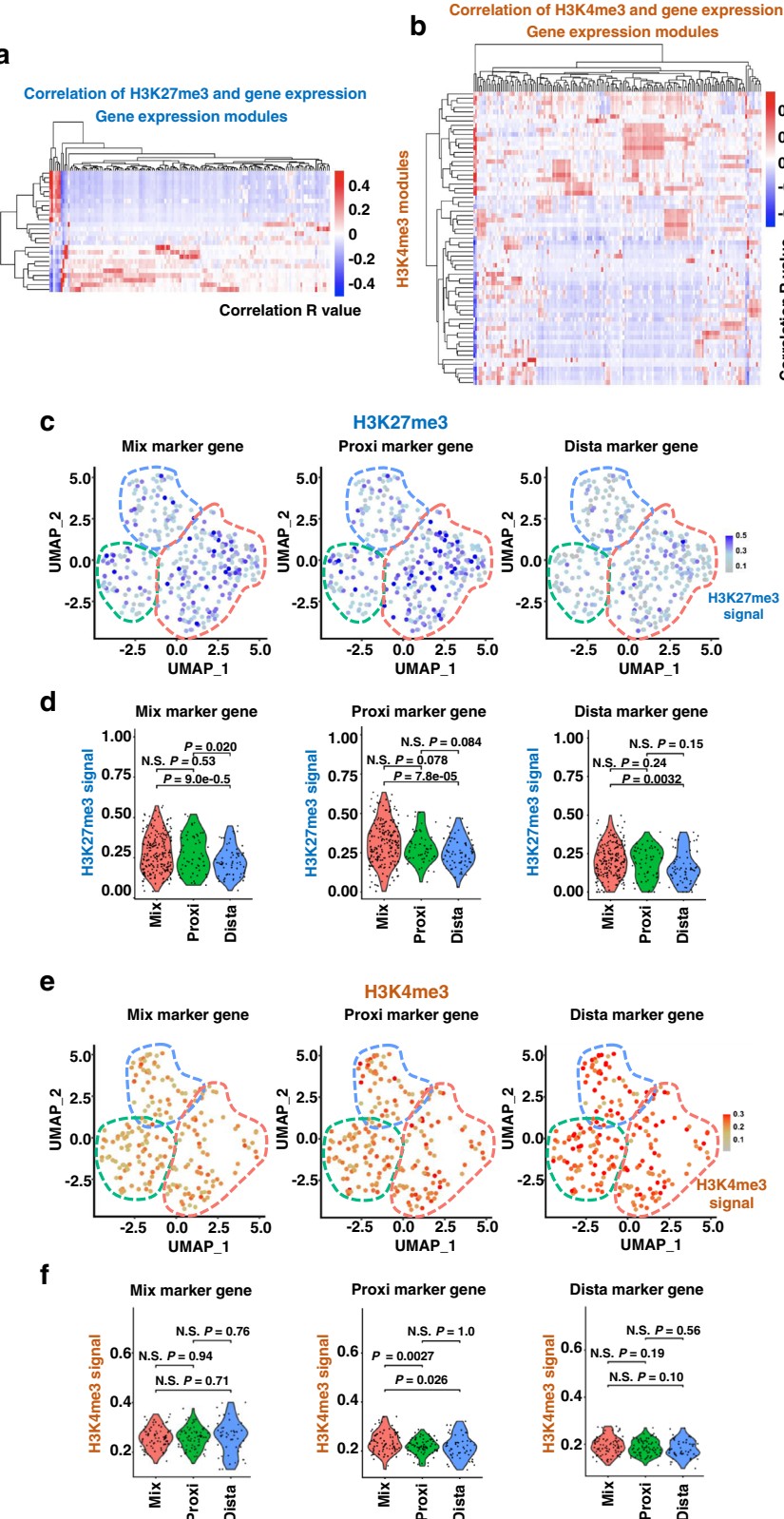

cluster cells at average expression levels, consistent with the observation of high H3K27me3 signals at marker genes in Mix cluster cells (Fig. 6a). Conversely, *Jarid2*, *Eed* and *Rbbp4* were expressed at high levels in cells from both the Proxi and Dista clusters. *Ezh2* and *Suz12* were upregulated in cells of the Dista cluster. Notably, *Rbbp7* was not detected using scSET-seq.

We then examined whether *Aebp2* was enriched at cluster marker genes using the GEO ChIP-seq dataset in mESCs GSE83082 [https://www.ncbi.nlm.nih.gov/geo/query/acc.cgi?acc=GSE83082]. AEBP2 ChIP-seq peaks were detected at the promoters of 200/344 marker genes. We utilized the CRISPR/Cas9 system to knock out *Ezh2* and *Aebp2* and to further

**Fig. 4 H3K27me3 is dynamically changed during mESC division with Wnt3a beads. a** Heatmap showing the correlations between H3K27me3 modules and transcriptional modules defined by WGCNA. Pearson correlation coefficient was calculated by module scores. **b** Heatmap showing the correlations between H3K4me3 modules and transcriptional modules defined by WGCNA. Pearson correlation coefficient was calculated by module scores. **c** Enrichments of H3K27me3 at cluster marker genes in individual cells of H3K27me3 scSET-seq. Total levels of H3K27me3 at the Mix, Proxi, and Dista cluster marker genes were shown respectively. **d** Violin plots showing the summarized signals in each cluster of cells. Cells were divided into Mix, Proxi, and Dista clusters and shown as the x-axis. H3K27me3 signals at all marker genes from Mix, Proxi, and Dista clusters were shown respectively. P values were calculated by Student's t-test, two-sided. N.S. Not Significant. $n = 208$ (H3K27me3 Mix cluster), 58 (H3K27me3 Proxi cluster), and 60 (H3K27me3 Dista cluster) cells. **e** Same as in **c**, except H3K4me3 scSET-seq was shown. **f** Same as in **d**, except H3K4me3 scSET-seq was shown. P values were calculated by Student's t-test, two-sided. N.S. Not Significant. $n = 83$ (H3K4me3 Mix cluster), 75 (H3K4me3 Proxi cluster), and 52 (H3K4me3 Dista cluster) cells. Source data are provided as a Source Data file.

test the functions of *Aebp2* in asymmetric cell division. Cell clones were generated from two different sgRNAs for each gene (Supplementary Fig. S14a and b). The total levels of H3K27me3 were inhibited in the *Ezh2* knockout (KO) clones (Fig. 6b). Accordingly, H3K27ac levels were slightly increased. *Aebp2* KO had no observable effects on the total levels of H3K27me3 or H3K27ac, as detected using Western blotting. Thereafter, we analyzed how *Ezh2* and *Abep2* KO clones were divided after treatment with Wnt3a beads (Fig. 6c). As previously reported[12,15], 62.2% of cells had asymmetrically divided, 24.5% of cells had symmetrically divided, and 13.3% of cells exhibited reversed division among the 233 analyzed wild-type cells, showing higher *Nanog* expression in distal cells (Fig. 6d). *Nanog* expression was analyzed in approximately 150 cells from each KO cell line. The ratios of symmetrically divided cells determined based on the *Nanog* signal[12,15,57] were decreased when *Aebp2* was knocked out. *Ezh2* KO had no obvious effect on the ratios of divided cells. Compared to parental cells, the ratios of reversed cells were not significantly altered in all KO cell lines. Other regulators, but not *Ezh2* or *Aebp2*, might have been responsible for the reversed division. These data suggest that *Aebp2* is important for symmetric cell division.

We conducted SET-seq to further identify genome-wide changes in gene expression and H3K27me3 levels in *Aebp2* KO cells. Two independent experiments were performed using each KO cell line. These two replicate analyses of gene expression and H3K27me3 correlated well (Supplementary Tables 1 and 2). In addition, the two *Aebp2* KO clones showed a consistent gene expression profile (Supplementary Fig. S14c). We then merged the gene expression profiles of the two *Aebp2* KO clones to obtain consistently changed genes, which were defined based on an absolute $\log_2(FoldChange) > 1$ and P value less than 0.05 (Supplementary Fig. S14d). A total of 1,414 genes were upregulated, and 1,403 genes were downregulated. Among the 321 marker genes, only 33 and 47 were up- and downregulated in *Aebp2* KO cells, respectively (Supplementary Fig. S14d and e), suggesting that the expression of cluster marker genes was not substantially altered. The pluripotency marker genes *Oct4*, *Nanog*, and *Sox2* were not changed following *Aebp2* KO.

The H3K27me3 sequencing results correlated well between the two KO cell lines (Supplementary Fig. S14f). We then merged the two cell lines and two replicates to detect consistent changes in H3K27me3 levels. We called 43,747 and 59,770 H3K27me3 peaks in WT and *Aebp2* KO cells, respectively. Although 30,332 peaks overlapped between WT and *Aebp2* KO cells, 29,438 peaks were unique to *Aebp2* KO cells (Supplementary Fig. S14g). We further analyzed the enrichment of H3K27me3 at peaks with or without overlapping AEBP2 peaks that were detected in NCBI Gene Expression Omnibus (GEO) AEBP2 ChIP-seq dataset GSE83082 [https://www.ncbi.nlm.nih.gov/geo/query/acc.cgi?acc=GSE83082]. H3K27me3 was present at higher levels at the H3K27me3 peaks that overlapped with AEBP2 peaks than at those that did not overlap (Supplementary Fig. S14h). More

importantly, in *Aebp2* KO cells, H3K27me3 levels increased at the peaks overlapping with AEBP2 peaks but did not change at the peaks without AEBP2 peaks. This result was consistent with previous reports showing that although AEBP2 increased the enzymatic activity of PRC2 in vitro, cells without AEBP2 exhibited elevated H3K27me3 levels at AEBP2 target sites[60,61]. This result is possibly due to the increased presence of other PRC2 subcomplexes or the formation of hybrid PRC2 subcomplexes at AEBP2-depleted loci, leading to increased PRC2 recruitment to target loci[62]. We then compared the enrichment of AEBP2 and H3K27me3 at cluster marker genes that were identified based on gene expression. Compared to Mix cluster marker genes, AEBP2 was present at high levels at Proxi and Dista cluster marker genes in wild-type cells (Fig. 6e). Similar to AEBP2, H3K27me3 levels were higher at marker genes of the Proxi and Dista clusters than at those of the Mix cluster. In addition, H3K27me3 levels increased at Mix cluster marker genes and were slightly decreased at Proxi cluster marker genes when *Aebp2* was knocked out (Fig. 6f). The compensation of increased H3K27me3 levels may explain the observation that not all daughter cells asymmetrically expressed *Nanog* following *Aebp2* KO.

## Discussion

Here, we report a strategy, SET-seq, for jointly profiling gene expression and histone modification in the same single cells. mRNA/cDNA hybrids can be tagmented by Tn5, probably because of the structural similarity between Tn5 and RNase H[46,47]. To utilize this feature for profiling gene expression, we thoroughly surveyed tagmentation conditions, including temperature, time, and enzyme concentration, to utilize this recently identified feature for profiling gene expression. Through these studies, we determined reproducible and time-saving conditions for the tagmentation of mRNA/cDNA hybrids. Furthermore, the constructed libraries with the same structures for the epigenome and transcriptome are convenient, which might simplify the downstream bioinformatics analysis. SET-seq is quite flexible for transcriptome profiling and thus the library can be constructed using any conventional strategy available in the laboratory, such as the powerful and widely used Smart-seq2[63]. In addition, the epigenomic profiling method can be improved with small modifications. For example, two or more chromatin elements can be recognized by corresponding antibodies and indexed using different transposomes for coprofiling. The other improvement would be that once the first tagmentation is performed, sequential ChIP could be conducted to map the concurrency of two chromatin binding proteins at the same loci.

The integrated scSET-seq analysis provides exclusive properties to map cell clusters and trajectories during cell differentiation and in heterogeneous populations. Similar to Smart-seq2, scSET-seq is suitable for the throughput of hundreds of cells. Compared to other single-cell-based methods, scSET-seq has a high gene recovery rate and positive epigenetic signals (Fig. 1e–h),

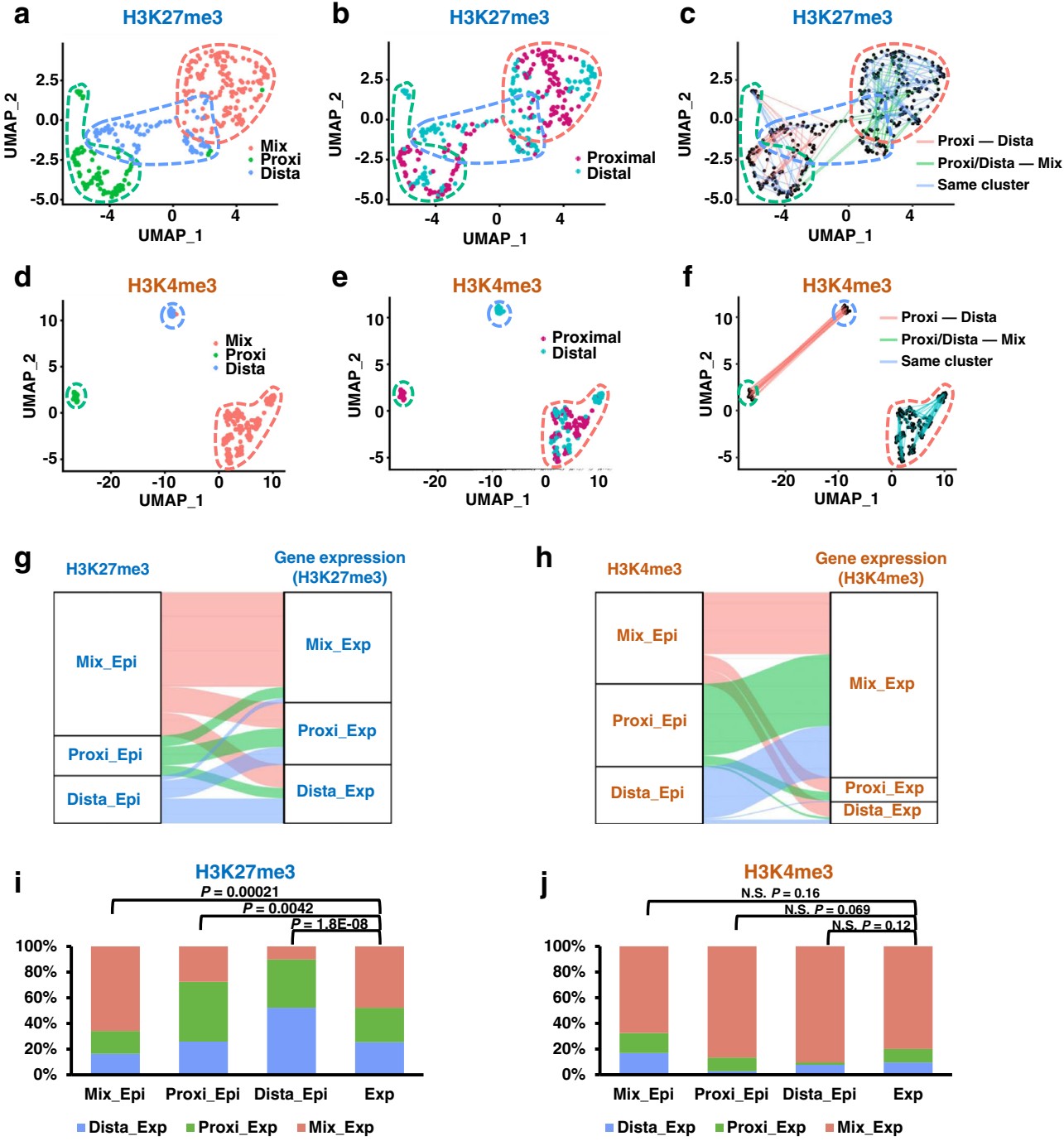

**Fig. 5 Cells clustered by H3K27me3 recapitulate their distributions clustered by gene expression. a** UMAP embedding showing the clustering of cells from H3K27me3 scSET-seq by H3K27me3 signals. Each dot represented a single cell. The cluster with a majority of proximal cells was defined as the Proxi cluster, with abundant distal cells was defined as the Dista cluster, and with mixed proximal and distal cells was defined as the Mix cluster. **b** Colored map showing the relative position of cells to Wnt3a beads in H3K27me3 scSET-seq. Cells proximal to beads were marked garnet and distal to cells were marked cyan. **c** Line maps showing the paired daughter cells among cell clusters in H3K27me3 scSET-seq. Red lines indicated the paired cells were between Proxi and Dista cluster, green lines indicated the two daughter cells were between Proxi/Dista cluster and Mix cluster, and blue lines connected cells inside the same cluster. **d** Same as in **a**, except H3K4me3 scSET-seq was analyzed. **e** Same as in **b**, except H3K4me3 scSET-seq was analyzed. **f** Same as in **c**, except H3K4me3 scSET-seq was analyzed. **g** Sankey plots showing the cell identities classified by H3K27me3 and joint transcriptional profiles. Epi, cell clusters defined by epigenetic profiles. Exp, cell clusters identified by gene expression profiles. **h** Sankey plots showing the cell identities classified by H3K4me3 and joint transcriptional profiles. **i** Bar graph showing the percentage of cells projected by gene expression profiles from H3K27me3 scSET-seq. Cell clusters identified by H3K27me3 were shown as the *x*-axis. Exp, the percentage of cells classified by corresponding gene expression was used as a negative control. *P* values were calculated by Student's *t*-test, two-sided. **j** Same as in **i**, except H3K4me3 scSET-seq data were analyzed. N.S. Not Significant. Source data are provided as a Source Data file.

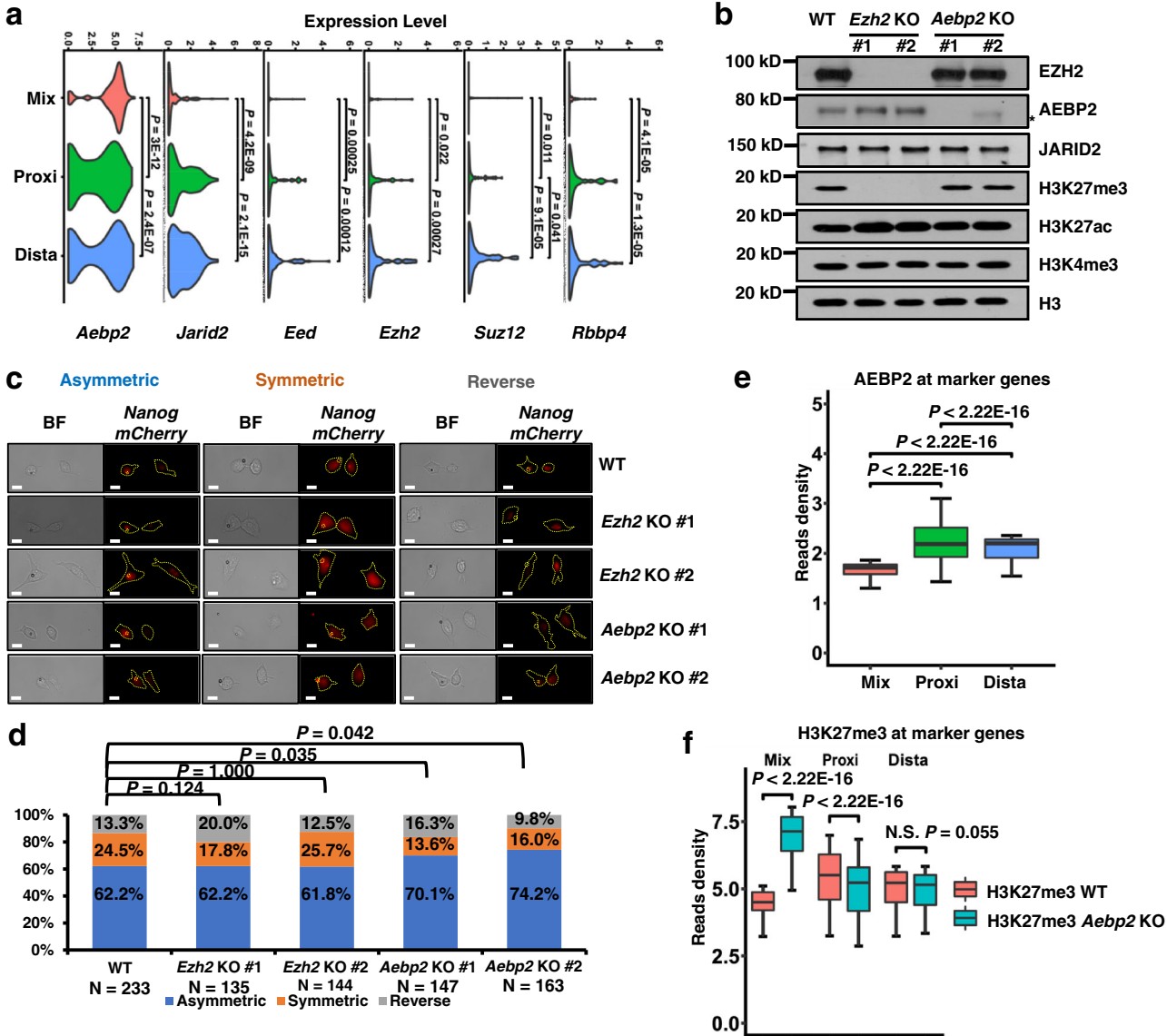

**Fig. 6 Aebp2 knockout increases the cells asymmetrically expressing Nanog-mCherry. a** Violin plots showing the average expression levels of components of PRC2 in the Mix, Proxi, and Dista clusters. To increase the reproducibility, the gene expressions of H3K27me3 and H3K4me3 scSET-seq were merged for plotting. *P* values were calculated by Student's *t*-test, two-sided. *Rbbp7* was not detected in the scSET-seq. *n* = 291 (Mix), 133 (Proxi), and 121 (Dista) cells. **b** Western blot showing the indicated protein levels in parental and KO mESCs. *Ezh2* KO decreased the total levels of H3K27me3 and increased the H3K27ac. *Aebp2* KO didn't exhibit obvious effects on the total levels of tested histone marks. Cell extracts were analyzed via Western blotting using the indicated antibodies, and two biological replicates were performed for each blot. Star indicated a nonspecific band. See Supplementary Fig. S14 for the Sanger sequencing results of KO clones. **c** Representative images of divided *Nanog-mCherry* mESCs cocultured with Wnt3a beads. Wnt3a beads and cells were marked in the immunofluorescence image. Asymmetric, higher *mCherry* amounts in the bead-proximal cell; Symmetric, similar amounts of *mCherry* in either cell; Reverse, higher amounts of *mCherry* in the bead-distal cell. BF, bright field. Scale bar, 15 μm. **d** The percentages of asymmetrically, symmetrically, and reversely divided cells, as defined by *Nanog-mCherry* signals, in parental, *Ezh2* and *Aebp2* KO cells. *P* values were calculated by chi-squared test. **e** The enrichments of AEBP2 at cluster marker genes that were identified by gene expression. AEBP2 peaks were first called and annotated to the closet transcription starting sites within 20 Kb. The reads density of Aebp2 was then calculated for peaks annotated to each gene. *P* values were calculated by Student's *t*-test, two-sided. The boxes were drawn from lower quartile (Q1) to upper quartile (Q3) with the middle line denoting the median, and whiskers with maximum 1.5 IQR (interquartile range). *n* = 90, 108, and 200 Aebp2 peaks at Mix, Proxi and Dista markers, respectively. *n* = 1000 (Mix), 1000 (Proxi), and 1000 (Dista) bins of Aebp2 at cluster marker genes. **f** The enrichments of H3K27me3 at cluster marker genes that were identified by gene expression. H3K27me3 peaks were first called in WT and *Aebp2* KO cells, merged, and annotated to the closet transcription starting sites within 20 Kb. The reads density of H3K27me3 was then calculated for peaks annotated to each gene. *P* values were calculated by Student's t-test, two-sided. The boxes were drawn from lower quartile (Q1) to upper quartile (Q3) with the middle line denoting the median, and whiskers with maximum 1.5 IQR (interquartile range). *n* = 504, 243, and 371 H3K27me3 peaks at Mix, Proxi, and Dista markers, respectively. Source data are provided as a Source Data file.

providing in-depth sequencing results for downstream analysis. When we called peaks in scSET-seq and compared them with bulk ENCODE data, much fewer peaks were called. A potential explanation is that we used stringent conditions for peak calling, and some low-quality peaks were excluded. Under this condition, approximately 60% of peaks in scSET-seq data overlapped with ENCODE peaks (Supplementary Fig. S6e and f). SET-seq, Paired-Tag and CoTECH are similar in terms of the preparation of epigenomic libraries but differ in the preparation of gene expression libraries. SET-seq probes cytoplasmic RNA with a large number of genes detected, while Paired-Tag identifies nuclear RNA with a medium number of genes recovered. In addition, SET-seq has a lower throughput than Paired-Tag and CoTECH. Thus, SET-seq is suitable for situations in which single cells cannot be sorted or selected using a high-throughput approach. In Wnt-induced asymmetric cell division, the target cells are distinguished based on their relative positions to Wnt3a beads. Therefore, conventional FACS sorting, which is based on fluorescence intensity, cannot be conducted for single-cell collection. Cells attached to beads or not are manually picked and subjected to scSET-seq for high recovery of epigenomic and transcriptional profiles.

We used cisTopic to remove potential noise before we incorporated epigenomic signals into single cells. The calculated signals are more specific in single cells. The throughput of this method is similar to Smart-seq2, which can be used to profile several hundred cells in one experiment. The number of detected cells is limited mainly by the asymmetric division system, as the current technology cannot determine the relative positions of cells to beads. Moreover, the FRiPs were slightly higher for H3K4me3 than for H3K27me3 in both scSET-seq and Paired-Tag analyses (Fig. 1h). Considering the noise of sequencing results, the accuracies of H3K27me3 would be lower than H3K4me3, and the decrease in correlations of H3K27m3 with gene expression would be larger than that of H3K4me3 with gene expression. Under this condition, we still found that H3K27me3 was highly correlated with changes in gene expression during asymmetric cell division.

Spatiotemporal cell fate determination is crucial for proper tissue organization, especially in stem cell niches, where Wnt signaling serves as a cue for stem cell self-renewal and cell fate determination in an evolutionarily conserved manner[7,64–67]. The effects of Wnt signaling on stem cells have been extensively analyzed when Wnt signaling is globally activated/inactivated. In vivo, Wnt signaling, which usually functions as a local gradient in stem cells, is understudied due to technical limitations. Recent technical progress in Wnt3a-coated beads and single-cell imaging have provided insights into Wnt-induced asymmetric cell division[12]. In addition to the finding that Wnt3a-coated beads function through the activation of Lrp6 and crosstalk with ionotropic glutamate receptor activity[57], newly synthesized and parental histones are segregated into daughter cells in a non-overlapping pattern[15], indicating an epigenomic alteration in asymmetric cell division. Asymmetric division of stem cells has been mainly elucidated in *Drosophila*, such as male germline stem cells (GSCs). Interestingly, almost all *Drosophila* male GSCs undergo asymmetric division to produce one GSC and one gonialblast after a cell cycle[68], while approximately 60% of mESCs divide asymmetrically when cells are attached to Wnt3a beads[12,15] (Fig. 6d). The heterogeneity of mESCs may be responsible for the different responses to the localized Wnt signal[69–71]. Previous studies are mainly based on high-resolution imaging to detect changes in gene expression and signaling activation. An informative approach would be to profile the epigenome and transcriptome of daughter cells for a mechanistic understanding. Using scSET-seq, several genes were classified as marker genes for asymmetrically divided cells, providing a list of

marker genes for future analyses of the molecular mechanisms of asymmetric cell division. In addition, the identification of the Mix cluster, which was branched from the main trajectory, provides other insights into asymmetric cell division, where the divided cells shifted to a different lineage other than the typical asymmetrically divided cells.

During the progression of differentiation, stem cells become restricted in their differentiation potential. Dynamic changes in histone modifications and associated chromatin structure are proposed to stabilize lineage-specific gene expression and regulate differentiation potential. Remarkably, the differentiation process might be reprogrammed through manipulation with small-molecule inhibitors or transcription factors that transform cells into a stem cell-like state or into different cell lineages. Chromatin assembly factor-1 (CAF-1), a histone chaperone responsible for the deposition of histone H3.1 to maintain chromatin structure, is discovered to function as a potent barrier in the reversion of pluripotent cells to a totipotent-like state, further implicating the essential chromatin structure in the regulation of cellular plasticity[72]. H3K27me3, which is usually linked to repressed gene expression, plays important roles in mammalian embryonic development and induced pluripotent stem cell (iPSC) generation. Generally, H3K4me3 is associated with active gene expression and regulates cell fate decisions. In mESCs, H3K27me3 and H3K4me3 are enriched at the same promoters as bivalent histone marks, which prime associated genes for rapid activation during development. Studies have shown drastic differences in the epigenomic profiles of hESCs and primary fibroblasts, whereas most changes arise from repressive histone marks, including H3K27me3[73]. Introduction of a heterozygous Y641F mutation in EZH2, which modifies the ratio of H3K27me2 and H3K27me3 in mESCs, is sufficient for the gain and suppression of cell lineage-specific gene expression and cellular phenotypes[74]. Moreover, the global loss of H3K27me3 marks facilitates the generation of iPSCs in mice and humans[75]. Similar to H3K27me3, repression of H3K4me3 improves the efficiency and blastocyst quality in somatic cell nuclear transfer[76]. Knockout of *Mll2*, which is responsible for the methylation of H3K4me3, results in impaired embryoid body formation and a failure to activate or delay in the activation kinetics of many bivalent genes that are key regulators of embryonic development and differentiation[77,78]. Mechanistically, MLL2 methylates H3K4 on many bivalent genes and protects these genes from repression by repelling PRC2 and the DNA methylation machinery[79]. During early embryonic development, the dynamic gain and loss of H3K4me3 and H3K27me3 occur at different embryogenesis stages. Cell context-dependent relationships among histone modifications are also important for the regulation of pluripotency and cell fate commitment, where different histone marks are dramatically changed at specific stages. When Wnt signaling is inhibited, mESCs are differentiated toward epiblast stem cell-like (EpiLC) states. Previous reports have shown that histone modifications are largely reorganized during the induction of mESCs to EpiLCs[80]. EpiLCs exhibit abundant bivalent gene promoters with decreased H3K27me3 levels compared to mESCs. Although EpiLCs are subsequently induced to differentiate into primordial germ cell-like cells (PGCLCs), H3K4me3 level initially decreases at the differentiation-related genes but subsequently increases with a concomitant increase in H3K27me3 levels. Consistent with the transformation from mESCs to EpiLCs, we also detected that mESCs treated without WNT3a showed substantial changes in H3K27me3 peaks, while H3K4me3 peaks largely overlapped (Supplementary Fig. S7c).

Using scSET-seq, we found that *Aebp2* KO increases the ratio of mESCs expressing *Nanog* asymmetrically with localized Wnt signaling. The core components of PRC2 in mammalian cells do

not have DNA binding capabilities but form complexes with regulatory proteins, including AEBP2 and JARID2[61,81–83]. Several lines of evidence have suggested that AEBP2 and JARID2 are critical for the enzymatic activities and recruitment of PRC2. JARID2, which mainly interacts with EED, is methylated by PRC2 and mimics a methylated H3 tail, whereas unmethylated AEBP2 mimics an unmethylated H3 tail and interacts with RBBP4 to stimulate PRC2 activity. AEBP2 is a conserved zinc finger protein that directly targets PRC2 to chromatin through its DNA binding ability[84]. In addition, JARID2 binds both DNA and long noncoding RNAs to recruit PRC2 onto chromatin[85,86]. Moreover, AEBP2 and JARID2 are assembled into the PRC2 complex together or independently, acting concurrently or individually in the regulation of PRC2 and subsequent H3K27me3 levels. In this way, AEBP2 and JARID2 function in a synergistic manner to promote enzymatic activity and to target PRC2 to specific chromatin loci, silencing key regulatory genes during the differentiation and development of mESCs[85,87]. Upon localized Wnt signaling, the average expression level of *Aebp2* is increased in symmetrically divided cells, but the average expression level of *Jarid2* is high in asymmetrically divided cells. The combined effects of *Aebp2* and *Jarid2* are not likely responsible for the cell fate decision. As previously reported[60,61], we also observed a minor increase in H3K27me3 levels at promoters when *Aebp2* was knocked out. AEBP2 is important for the formation of distinct PRC2 subcomplexes. Knockout of *Aebp2*, but not *Ezh2*, increases the ratio of mESCs expressing *Nanog* asymmetrically following localized Wnt signaling. The delicate and localized programming of H3K27me3, as determined by AEBP2, is needed during asymmetric cell division. In the Mix cluster, *Aebp2* increased H3K27me3 levels at marker genes to repress the activation of genes that were important for asymmetrical cell division. An interesting approach would to investigate how the changes in epigenome information at subsets of chromatin loci, but not the total levels, participate in the regulation of cell differentiation. The prolonged KO of *Aebp2* might also exert a secondary effect. A rescue or acute KO assay may help to address this possibility. In the rescue assay, *Aebp2* must be expressed at the original level, which is important to maintain cell fate determination. Acute KO timing may affect the KO effects. If *Aebp2* is knocked out too early, the prolonged loss of Aebp2 may lead to a secondary effect. If *Aebp2* KO is induced too late in single cells, the AEBP2 protein is not fully degraded. These assays, if performed perfectly, might provide other clues to dissect the effect of *Aebp2* on cell fate determination.

Several improvements will help us to better understand the cell fate decision during Wnt3a-induced asymmetric cell division. We monitored the division of parental cells with white light to confirm that two daughter cells were derived from the same parental cell. The intensities of *Nanog* were not recorded since we needed to proceed the cells within a short time. Cells migrated fast, and many of the two paired daughter cells moved to a slightly different distance in the Z axis, leading to the loss of imaging focus for one of the cells when immunofluorescence images were captured. The collection of a sufficient number of cells for scSET-seq is very difficult if the immunofluorescence intensities must be recorded in all cells. Studies recording this information for the comparison of cell clusters and immunofluorescence intensities in further experiments would be interesting. During Wnt3a bead-induced asymmetric cell division, a small population of cells divided in the opposite direction, exhibiting higher *Nanog* signals in distal cells than in proximal cells. How this population of cells is regulated is also interesting. Because of their rarities, we did not obtain a sufficient number of cells to determine the epigenome and transcriptome profiles of these cells using scSET-seq. The technologies that can be used to sort the cells based on their relative positions to beads should be developed to increase the throughput for scSET-seq. In addition, with modifications of pooling the samples before library construction, this method can be improved to increase the throughput and reduce the cost. In the future, an interesting approach would be to use this system to analyze how parental and newly synthesized histones are inherited during asymmetric cell division. These analyses would provide more evidence to understand the molecular mechanisms underlying asymmetric cell division.

## Methods

**mESCs culture and incubation with Wnt3a beads**. Mouse embryonic stem cell ES-E14TG2a from ATCC (CRL-1821) were cultured on 0.1% gelatin-coated plates in DMEM medium containing 15% Fetal Bovine Serum, 1% Penicillin/Streptomycin, 1% Glutamax, 0.1 mM 2-mercaptoethanol, 1% MEM Non-Essential Amino Acids, 1% Sodium Pyruvate, and 1,000 U/ml recombinant leukemia inhibitory factor (LIF).

Wnt3a coated beads were prepared as described[88]. Localized Wnt3a beads with mESCs were engineered in the 2 cm plates following the previous method[15]. To achieve one cell and one bead contact, mESCs were seeded at a low concentration before imaging or collecting. After seeding the cells and Wnt3a beads, samples were collected at around 12 h.

**Primers**. Primers used in this study were listed in Supplementary Dataset 6.

**RT-PCR**. RNA was isolated by UNlQ-10 Column Trizol Total RNA Isolation Kit (Sangon Biotech, Cat.# B511321) according to manufacturer instructions. 500 ng of total RNA was used for reverse transcription by EasyScript® One-Step gDNA Removal and cDNA Synthesis SuperMix (TransGen, Cat.# AE311-02) according to manufacturer instructions. Two-step qPCR (95 °C for 30 s; 40 cycles of 95 °C for 5 s and 60 °C for15 s) was performed in 10 μl reaction with 0.1 μl 10 mM RT-PCR primers (Supplementary Dataset 6) and 5 μl Hieff qPCR SYBR Green Master Mix (YEASEN, Cat.# 11201ES08) using Quantagene q225 qPCR system (Kubo Technology, Beijing). β-actin was used as a control to normalize the expression of other genes.

**pA-Tn5 protein purification**. Plasmid expressing pA-Tn5 was constructed from vector pTXB1. pA-Tn5 was purified as described before with minor modifications[89]. Briefly, cells expressing pA-Tn5 were resuspended in 40 ml chilled cell lysis buffer (20 mM HEPES pH 7.2, 0.8 M NaCl, 1 mM EDTA, 10% glycerol, 0.1% Triton X-100). The lysate was sonicated for 4 times (15 s on and 15 s off) on ice and then centrifuged at 30,970 × g, 4 °C for 30 min. 500 μl 10% PEI was added drop by drop with stirring. The precipitate was removed by centrifugation at 17,420 × g, 4 °C for 15 min. Cleared supernatant was incubated with 2 ml pre-washed IgG-Sepharose beads (washed with 10 ml cell lysis buffer for three times) for 6 h. Then the beads were washed with 10 ml Washing buffer (20 mM HEPES at pH 7.2, 0.3 M NaCl, 1 mM EDTA, 10% glycerol 0.1% Triton X-100) for four times. pA-Tn5 was eluted with 2 ml 0.5 M NH₄Ac (pH 3.0) twice and then neutralized by 1 M Tris-HCl (pH 9.0) to pH 7.2 immediately. Protein was dialyzed twice to dialysis buffer (40 mM HEPES pH 7.2, 0.2 M NaCl, 0.2 mM EDTA, 2 mM DTT, 20% glycerol) at 4 °C.

**Transposome assembly**. Transposomes were assembled following the protocol as described before[90]. To anneal adapters, Tn5ME-A and Tn5ME-B oligos at 100 μM (I5_transposome and I7_transposome oligos for indexed transposome assembly) was mixed with equal volume of Tn5MErev oligos at 100 μM respectively (Supplementary Dataset 6). Oligos were placed in a thermal cycler at 95 °C for 5 min followed by gradually cooling at 0.1 °C/s to 25 °C. Then equal amount of annealed Tn5ME-A and Tn5ME-B (I5_transposome and I7_transposome for indexed transposome assembly) oligos was mixed. 70 μl pA-Tn5 protein at around 0.5 mg/ml was incubated with 10 μl of mixed primers at room temperature for 1 h.

**In vitro mRNA/cDNA hybrid tagmentation and sequencing**. Total RNA was extracted by the UNIQ-10 Column Total RNA Purification kit (Sangon Biotech). Genomic DNA was digested by 0.5 μl DNase I (NEB) at 37 °C for 30 min and then inactivated at 75 °C for 10 min. mRNA/cDNA hybrids were reverse transcribed by Thermo Fisher SuperScript™ IV kit under the manufacture's instruction. Samples were tagemented in DMF buffer (10% DMF, 10 mM Tris-HCl pH 7.5, 2 mM MgCl₂) by Tn5 transposomes at different temperature and time as indicated in a 20 μl reaction. To stop the reaction and release the tagmented cDNA, 2.25 μl 0.5 M EDTA, 2.75 μl 10% SDS, and 0.5 μl 20 mg/ml Proteinase K were added to the 20 μl reaction and incubated at 55 °C for 30 min followed by 70 °C for 20 min. DNA was purified by 1.2X AMPure XP beads. Library preparation was done as described previously[91]. In brief, 20 μl purified DNA was mixed with 2.5 μl of uniquely barcoded P5_nextera and P7_nextera primer (10 μM) (Supplementary Dataset 6). 25 μl NEBNext HiFi 2X PCR Master mix was added and mixed.

Samples were placed in a thermo cycler using the following cycling condition: 72 °C for 5 min; 98 °C for 30 s; 15 cycles of 98 °C for 10 s and 63 °C for 10 s; final extension at 72 °C for 1 min and hold at 8 °C. Post-PCR clean-up was performed by adding 1.2X volume of AMPure XP beads. Libraries were sequenced in paired end 150 bp mode on the NovaSeq platform.

**Ligation based RNA-seq.** Ligation based RNA-seq libraries were prepared as described before[92]. In brief, the poly-A mRNA was purified from the total RNA of mESCs by poly-T oligo. Then, the poly-A mRNA was fragmented and reverse transcribed into cDNA with random primers. A second strand cDNA was synthesized with DNA Polymerase I and RNase H. The single A base was added for the ligation of the cDNA to sequencing primers. The cDNAs were amplified for sequencing with pair-end reads of 100 bp on the BGISEQ-500 platform.

**Bulk SET-seq.** Cells were harvested and then lysed in 1 ml NE buffer (20 mM HEPES pH 7.5, 0.5 mM KCl, 0.5 mM Spermidine, 0.5% Triton X-100, 20% Glycerol, 1 mM PMSF) on ice for 10 min. 10 μl Concanavalin A coated magnetic beads (Bangs Laboratories, Inc) were washed twice with Binding buffer (20 mM HEPES pH 7.5, 0.5 mM KCl, 0.5 mM Spermidine, 0.5% Triton X-100, 20% Glycerol, 2 mM MnCl$_2$ 1 mM PMSF) before being added to each sample. Samples with beads were then incubated at room temperature for 10 min.

Beads-bound nuclei were treated as CUT&Tag with modifications[40]. Samples were blocked with 1 ml Blocking buffer (20 mM HEPES pH 7.5, 150 mM NaCl, 0.5 mM Spermidine, 0.5% BSA, 1 mM EDTA, 1 mM PMSF) at room temperature for 10 min. Samples were then incubated with 1:200 diluted anti-H3K27me3 or anti-H3K4me3 primary antibody in 50 μl Washing buffer (20 mM HEPES pH 7.5, 150 mM NaCl, 0.5 mM Spermidine, 0.5% BSA, 1 mM PMSF) at room temperature for 2 h or at 4 °C overnight. The nuclei were washed with Washing buffer twice (20 mM HEPES pH 7.5, 150 mM NaCl, 0.5 mM Spermidine, 0.5% BSA, 1 mM PMSF) before incubated with 1:100 diluted anti-rabbit secondary antibody at room temperature for 30 min. After washing with Washing buffer (20 mM HEPES pH 7.5, 150 mM NaCl, 0.5 mM Spermidine, 0.5% BSA, 1 mM PMSF) once, 50 μl of pA-Tn5 binding reaction was conducted at room temperature for 1 h by 1:25 dilution of assembled pA-Tn5 transposomes in Washing buffer (20 mM HEPES pH 7.5, 150 mM NaCl, 0.5 mM Spermidine, 0.5% BSA, 1 mM PMSF). Samples were washed twice with Washing buffer (20 mM HEPES pH 7.5, 150 mM NaCl, 0.5 mM Spermidine, 0.5% BSA, 1 mM PMSF). Tagmentation was initiated by adding 50 μl tagmentation buffer (Washing buffer with 300 mM NaCl and 10 mM MgCl$_2$) and incubating at 37 °C for 1 h. To stop the tagmentation and release DNA, 2.25 μl 0.5 M EDTA, 2.75 μl 10% SDS, and 0.5 μl 20 mg/ml Proteinase K were added to the sample and incubated at 55 °C for 30 min followed by 70 °C for 20 min. DNA was purified by QIAquick PCR purification kit. Libraries were constructed using indexed P5_nextera and P7_nextera primers (Supplementary Dataset 6) as described above in mRNA/cDNA hybrid tagmentation and sequencing.

The supernatant after Concanavalin A beads binding was transferred to another RNase-free tube followed by centrifugation at 3,000 x g, 4 °C for 5 min to remove the unbound nuclei carryover. The supernatant was carefully transferred to another RNase free PCR tube and incubated with 1 μl DNase I (NEB) and 1 μl Recombinant RNase Inhibitor (Takara Bio) at 37 °C for 30 min followed by 70 °C for 10 min to further eliminate the genomic DNA carryover. The reverse transcription was done by using Thermo Fisher SuperScript™ IV kit. 11 μl sample was mixed with 1 μl 50 μM Oligo d(T)$_{20}$ and 1 μl 10 mM dNTP, incubated at 65 °C for 5 min, and then cooled on ice for 2 min immediately. The sample was mixed with 4 μl 5X SSIV Buffer, 1 μl 100 mM DTT, 1 μl Ribonuclease Inhibitor, and 1 μl SuperScript™ Reverse Transcriptase. Reactions were incubated at 42 °C for 90 min, 10 cycles of 50 °C for 2 min followed by 42 °C for 2 min, and 85 °C for 5 min. mRNA/cDNA hybrids were then tagmentated in DMF buffer (10% DMF, 10 mM Tris-HCl pH 7.5, 2 mM MgCl$_2$) with 1 μg pA-Tn5 transposomes at 37 °C for 5 min in a 20 μl reaction. Tagmentation was stopped by adding 2.25 μl 0.5 M EDTA, 2.75 μl 10% SDS, and 0.5 μl 20 mg/ml Proteinase K to every 20 μl reaction. Samples were incubated at 55 °C for 30 min followed by 70 °C for 20 min. DNA was purified by 1.2X AMPure XP beads for library construction using indexed P5_nextera and P7_nextera primers (Supplementary Dataset 6) as described above.

**scSET-seq.** Four hours after seeding the mESCs with Wnt3a beads, cell culture plates were mounted in live cell image system (DeltaVision Elite Deconvolution Microscope). Cells connected to one Wnt3a beads were monitored until they were collected for scSET-seq (Supplementary Movie 1). mESCs were washed with PBS once and slightly trypsinized for 3 min. The trypsinization was stopped by adding fresh medium. Single cells were picked by mouth pipette into the PBS with 0.04% BSA drops. Until the cells were completely dispersed, single cells were transferred under the stereoscopic microscope into 96 well-plate with 8 μl NE Buffer (20 mM HEPES pH 7.5, 0.5 mM KCl, 0.5 mM Spermidine, 0.5% Triton X-100, 20% Glycerol) plus 1:20 diluted Recombinant RNase Inhibitor (Takara Bio). Concanavalin A coated magnetic beads were washed twice with Binding buffer (20 mM HEPES pH 7.5, 0.5 mM KCl, 0.5 mM Spermidine, 0.5% Triton X-100, 20% Glycerol, 2 mM MnCl$_2$ 1 mM PMSF). 1 μl pre-washed Concanavalin A coated magnetic beads were added to each well, and incubated at room temperature for 10 min. The

supernatant was transferred to another RNase-free 96-well plate with 0.25 μl DNase I (NEB) and 0.25 μl Recombinant RNase Inhibitor (Takara Bio) in each well.

Single nuclei were resuspended in 100 μl Blocking buffer (20 mM HEPES pH 7.5, 150 mM NaCl, 0.5 mM Spermidine, 0.5% BSA, 1 mM EDTA, 1 mM PMSF) at room temperature for 10 min. The supernatant was discarded and nuclei were resuspended in 10 μl Washing buffer (20 mM HEPES pH 7.5, 150 mM NaCl, 0.5 mM Spermidine, 0.5% BSA, 1 mM PMSF) with 1:200 diluted anti-H3K27me3 or anti-H3K4me3 primary antibody. Samples were incubated at 4 °C overnight. After washing with Washing buffer once (20 mM HEPES pH 7.5, 150 mM NaCl, 0.5 mM Spermidine, 0.5% BSA, 1 mM PMSF), 1:100 diluted anti-rabbit secondary antibodies were added in 10 μl Washing buffer and incubated at room temperature for 1 h. Samples were then washed once with Washing buffer (20 mM HEPES pH 7.5, 150 mM NaCl, 0.5 mM Spermidine, 0.5% BSA, 1 mM PMSF) and mixed with 1:25 diluted pA-Tn5 transposomes with indexed i5 and i7 primers (Supplementary Dataset 6) in 10 μl Washing buffer. After being incubated at room temperature for 1 h, genomic DNA was tagmented in 10 μl Tagmentation buffer (Washing buffer with 300 mM NaCl and 10 mM MgCl$_2$) at 37 °C for 1 h. The reactions were stopped by adding 1.12 μl 0.25 M EDTA, 1.37 μl 10% SDS, and 0.25 μl 20 mg/ml Proteinase K. Samples were then pooled together. DNA was purified by QIAGEN MinElute PCR Purification Kit.

The supernatant after Concanavalin A beads binding was used for gene expression profiling. After mixed with 0.2 μl DNase I (NEB) and 0.2 μl Recombinant RNase Inhibitor (Takara Bio), the supernatant was incubated at 37 °C for 30 min followed by 70 °C for 10 min. Each sample was mixed with 1 μl 50 μM Oligo d(T)$_{20}$ and 1 μl 10 mM dNTP, incubated at 65 °C for 5 min, and immediately put on ice for 2 min. Reverse transcription was conducted by mixing with 4 μl 5X SSIV Buffer, 1 μl 100 mM DTT, 1 μl Ribonuclease inhibitor, and 0.2 μl SuperScript™ Reverse Transcriptase. Samples were incubated at 42 °C for 90 min; 10 cycles of 50 °C for 2 min and 42 °C for 2 min; 85 °C for 5 min. Reverse transcribed mRNA/cDNA hybrids were then tagementated by 0.5 μg transposomes with indexed i5 and i7 (Supplementary Dataset 6) in DMF buffer (10% DMF, 10 mM Tris-HCl pH 7.5, 2 mM MgCl$_2$) at 37 °C for 5 min in a 20 μl reaction. Reaction was stopped by adding 2.25 μl 0.5 M EDTA, 2.75 μl 10% SDS, and 0.5 μl 20 mg/mL Proteinase K and incubated at 55 °C for 30 min followed by 70 °C for 20 min. Samples were pooled and purified by QIAGEN MinElute PCR Purification Kit to 30 μl H$_2$O.

Libraries from epigenome and transcriptome were constructed with a structure similar to TruSeq libraries for pooling with other samples as described before[51]. 20 μl eluted DNA was mixed with 2 μl connect forward primer and connect reverse primer (10 μM) (Supplementary Dataset 6). 25 μl NEBNext HiFi 2X PCR Master mix was added and mixed. Samples were amplified using the following cycling condition: 72 °C for 5 min; 95 °C for 30 s; 15 cycles of 98 °C for 10 s, 63 °C for 30 s, and 72 °C for 1 min; 72 °C for 5 min, and hold at 8 °C. 1 μl Exonuclease I (NEB) was added and incubated at 37 °C for 30 min followed by 80 °C for 20 min. Samples were then mixed with 1.25 μl Universal Primer (10 μM) and 1.25 μl P7_TruSeq index primer (10 μM) (Supplementary Dataset 6), 2.5 μl H$_2$O, and 5 μl NEBNext HiFi 2X PCR Master mix. Amplification was conducted in a thermo cycler using the following cycling condition: 95 °C for 30 s; 8 cycles of 98 °C for 10 s, 63 °C for 30 s, and 72 °C for 1 min; 72 °C for 5 min, and hold at 8 °C. Post-PCR clean-up was performed by adding 0.5X volume of AMPure XP beads, incubating at room temperature for 10 min, removing beads by magnet stand, and adding 0.5X volume of AMPure XP beads again. Samples with beads were incubated at room temperature for 10 min, washed twice by 80% ethanol, and eluted in 20 μl H$_2$O. Libraries were sequenced in paired end 150 bp mode on the NovaSeq platform.

**Antibodies.** Rabbit polyclonal anti-Histone H3 (Cat.# ab1791, Abcam), 1:5000 diluted for Western blot; Rabbit polyclonal anti-Histone H3K27me3 (Cat.# 9733, Cell Signaling Technology), 1:1000 diluted for Western blot, 1:50 diluted for SET-seq, 1:200 diluted for scSET-seq; Rabbit polyclonal anti-Histone H3K4me3 (Cat.# ab8580, Abcam), 1:1000 diluted for Western blot, 1:50 diluted for SET-seq, 1:200 diluted for scSET-seq; Rabbit polyclonal anti-Histone H3K9me3 (Cat.# ab8898, Abcam), 1:1000 diluted for Western blot; Rabbit polyclonal anti-Histone H3K27ac (Cat.# ab4729, Abcam), 1:1000 diluted for Western blot; Rabbit monoclonal anti-EZH2 (Cat.# 5246, Clone name D2C9, Cell Signaling Technology), 1:1000 diluted for Western blot; Rabbit monoclonal anti-AEBP2 (Cat.# 14219, Clone name D7C6X, Cell Signaling Technology), 1:1000 diluted for Western blot; Rabbit polyclonal anti-JARID2 (Cat.# G-2, Novus Biologicals), 1:1000 diluted for Western blot; Peroxidase AffiniPure Goat anti-Rabbit IgG (H + L) (Cat.# 111-035-003, Jackson ImmunoResearch Laboratories), 1:1000 diluted for Western blot.

**Bulk transcriptional SET-seq data processing.** Sequencing reads adapter and low-quality bases were removed by Trim Galore (version 0.6.4) [https://www.bioinformatics.babraham.ac.uk/projects/trim_galore] with the parameter '--paired'. Filtered reads were mapped to the mouse reference UCSC mm9 [https://hgdownload.soe.ucsc.edu/goldenPath/mm9/bigZips/], which excluded mitochondria and random chromosomes, by STAR (version 2.5.4b)[93]. Expression matrixes were counted by featureCounts[94] (version 2.0.0) and normalized as reads per kilobase per million mapped reads (RPKM) by R package edgeR (version 3.28.1)[95]. The differential analysis was done by R package DEseq2[96] (version 1.26.0). To calculate the correlation of different RNA-seq samples, ellipse correlation plots

were plotted by R package corrplot (version 0.84) [https://github.com/taiyun/corrplot] after merging two repeats. Heatmaps were drawn by R package pheatmap (version 1.0.12) [https://github.com/raivokolde/pheatmap] using top 2000 average expression genes. The Venn plots were plotted by R package VennDiagram (version 1.6.20) [https://github.com/cran/VennDiagram]. The number of reads mapped to exons or introns was calculated by Homer[96] (version 4.11) with the parameter 'analyzeRepeats.Pl -count exons -raw' or 'analyzeRepeats.pl -count introns -raw'. The correlations between two repeats were summarized in Supplementary Table 1. The correlation was calculated by the coefficient of determination (R^2) of their shared genes' RPKM values. The number of total reads and unique mapped reads were summarized in Supplementary Dataset 7.

**Receiver operating characteristic (ROC) curve**. ROC curves were used to evaluate the qualities of bulk SET-seq of histone mark with different cell numbers[49]. Particularly, peaks recovered in ENCODE H3K27me3 ChIP-seq ENCSR059MBO [https://www.encodeproject.org/experiments/ENCSR059MBO/] at promoter regions (3,000 bp downstream to 500 bp upstream of gene TSS) were used as standard positives and promoter regions not overlapped with ENCODE H3K27me3 ChIP-seq peaks were defined as standard negatives. Following the standard, ROC curves were constructed by the following definition: true positives (TPs), peaks that overlapped with standard positives; false positives (FPs), peaks that did not overlap with standard positives; false negatives (FNs), peaks that did not overlap with standard negatives; true negatives (TNs), peaks that overlapped with standard negatives. The true positive rate (TPR) was defined as $TP/(TP + FN)$ and the false positive rate (FPR) was defined as $FP/(FP + TN)$. TPR and FPR value sets were used to construct ROC curves by varying the peak-calling threshold.

**Bulk epigenomic SET-seq data processing and peak calling**. Adapters and low-quality bases in data were removed by Trim Galore (version 0.6.4) [https://www.bioinformatics.babraham.ac.uk/projects/trim_galore] with the parameter '--paired' after merging two repeats. Trimmed reads were then mapped to the mouse reference genome UCSC mm9 [https://hgdownload.soe.ucsc.edu/goldenPath/mm9/bigZips/] using Bowtie2[97] (version 2.3.5.1). PCR duplicates were removed by GATK (version 4.1.4.0) [https://github.com/broadinstitute/gatk] with the parameter '--REMOVE_DUPLICATES = true'. The reads mapped to mitochondria and random chromosomes were filtered out. Peaks were identified using MACS2 (version 2.2.6)[98] to call broad peaks for H3K27me3 and narrow peaks for H3K4me3. ENCODE H3K27me3 ChIP-seq and its input data were downloaded from ENCSR059MBO [https://www.encodeproject.org/experiments/ENCSR059MBO/] and ENCSR326ULS [https://www.encodeproject.org/experiments/ENCSR326ULS/] as reference data. Peaks overlapped among different samples were generated by Homer with the parameter 'mergePeaks –venn -d 3000'. Peak structures of different concentrations were generated by R package ChIPseeker[99] (version 1.22.1).

BEDTools[100] (version 2.92.2) and bedGraphToBigWig (version 4) [https://www.encodeproject.org/software/bedgraphtobigwig/] were used to normalize mapped reads and calculate the coverage of signals with the following parameters 'genomecov --scaleFactor 10,000,000/mapped_reads_number'. Heatmaps were generated by computeMatrix and plotHeatmap with reference-point mode in deepTools[101] (version 3.4.3). Library complexity was estimated and predicted using the preseq[102] (version 2.0.3). The number of total reads and unique mapped reads were summarized in Supplementary Dataset 7. The correlations between two repeats were summarized in Supplementary Table 2. A 1 Kb sliding window across the whole genome was used to calculate the Pearson product moment correlation. Peak qualities were summarized in Supplementary Dataset 8.

**Transcriptional scSET-seq data processing**. To get the single-cell gene expression matrixes, pipeline tool zUMIs[103] (version 2.7.1c) was used to split reads into each cell by cell-specific barcode and map reads to mouse genome UCSC mm9 [https://hgdownload.soe.ucsc.edu/goldenPath/mm9/bigZips/], which excluded mitochondria and random chromosomes. The expression matrixes were analyzed by R package Seurat[53] (version 3.2.1) using genes mapped to both exons and introns. To compare the transcriptional scSET-seq with other conventional scRNA-seq data, Smart-seq2 data were downloaded from GSE151334 [https://www.ncbi.nlm.nih.gov/geo/query/acc.cgi?acc=GSE151334] and 48 cells were randomly picked from this data after quality control (filtering cells with detected gene numbers less than 1000). Saturation curves were constructed with detected gene numbers when reads were randomly gradient sampled 4 M, 3 M, 2 M, 1 M, 0.5 M, and 0.25 M from total reads.

**Epigenomic scSET-seq data processing and quality check**. Epigenomic scSET-seq data were de-multiplexed by in-house script by the combination of cell-specific barcodes in read1 and read2. The single-cell data were processed analogously as the bulk SET-seq data described above. To visualize scSET-seq signals, deepTools was used to normalize mapped reads and calculate the coverage of genome-wide continuous 50 bins with the following parameters 'bamCoverage --scaleFactor 10,000,000/mapped_reads_number' and signals were visualized in track view using

Integrative Genomics Viewer (IGV)[104] (version 2.6.3). To evaluate the performance of scSET-seq data, FRiPs was defined as the fraction of reads in peaks that called from merged single cell data as the positive reference peaks. scATAC-seq downloaded from GSE100033 [https://www.ncbi.nlm.nih.gov/geo/query/acc.cgi?acc=GSE100033] were used for comparisons with scSET-seq data.

**Embedding epigenomic and transcriptional data in bulk SET-seq**. deepTools was used to calculate the normalized signal scores by 'multiBigwigSummary BED-file' with SET-seq peak files from 10,000 cells as a reference. To define the enrichments of epigenomic signals to individual genes, peaks were annotated to nearby genes by R package ChIPseeker. Then signal scores were calculated at each gene and further normalized by peak length. H3K4me3 and H3K27me3 signal scores matrixes were merged according to their shared genes. Finally, epigenomic and transcriptional data were merged by both detected genes. Genes were clustered by hierarchical clustering algorithms using row scaled signal scores and heatmaps were plotted with column scaled signal scores by pheatmap.

**scSET-Seq data analysis for asymmetric cell division**. Transcriptional scSET-seq raw reads processing was performed as described above. R package Seurat was used for analyzing transcriptional data. Quality control was done by removing cells with less than 1,000 or more than 5,000 genes detected. To correct batch effect, canonical correlation analysis (CCA) was performed with Seurat function 'IntegrateData'. Datasets from H3K27me3 and H3K4me3 scSET-seq were aggregated for better reproduction of dimensional reduction. Differential genes were chosen by threshold (p value < 0.05 and the absolute value of $\log_2(FoldChange) > 2$) for linear dimensional reduction and nonlinear dimension. Clustering was performed by a shared nearest neighbor (SNN) algorithm with the function 'FindClusters'. Marker genes of each cluster were generated by function 'FindAllMarkers'. GO analysis was performed by clusterProfiler[105] (version 3.14.3) using all marker genes of each cluster. Marker genes ranked top 50 and top5 of each cluster were filtered according to $\log_2(FoldChange)$ to generate single-cell clustering plots and trajectory plots combined with epigenomic scSET-seq and visualize the gene expression distribution, respectively. For pseudotime analysis, differential genes ($P$ value <0.05 and the absolute value of $\log_2(FoldChange) > 2$) were selected to order cells in pseudotime by Monocle2[106] (version 2.14) and STREAM[58] (version 1.0). The function 'plot_genes_branched_heatmap' in Monocle2 was used to visualize the dynamics of differentially expressed genes. Violin plots of gene expression level were generated by function 'Vlnplot' in Seurat. The significance of gene expression across the three clusters was calculated by Student's t-test.

Epigenomic data from scSET-seq were processed as follows. To get normalized signature scores, cisTopic[59] (version 0.3.0) was used with default parameters. Aggregated cell peaks were used for counting reads and bulk SET-seq peaks from 10,000 cells were defined as signature regions. Peaks were preferably annotated to genes within 3 Kb from the transcription starting site, then to the closest gene within 500 Kb, as previously described[107]. To evaluate the correlation of scSET-seq data, kME scores were computed by WGCNA[108,109] (version 1.69) and gene modules were generated for epigenomic and transcriptional SET-seq respectively. The Pearson correlation coefficient matrix which was showed as a heatmap was constructed using module scores. The chi-square test was used to identify the significance between epigenomic and transcriptional modules from the same SET-seq. Signature scores were used for downstream clustering and pseudotime analysis. The top 2000 differential signature genes ranked by p value were used for analysis with Seurat and Monocle2. To evaluate the clustering result, ggalluvial [https://github.com/corybrunson/ggalluvial] (version 0.12.3) was used to show the changes of cell number between epigenomic and transcriptional clusters. The chi-square test was used to identify the significance between observed cell number distributions.

**Reporting summary**. Further information on research design is available in the Nature Research Reporting Summary linked to this article.

## Data availability

The reference genome used in this study is available in UCSC [http://genome.ucsc.edu] database under mouse reference genome mm9 [https://hgdownload.soe.ucsc.edu/goldenPath/mm9/bigZips/]. The raw and processed sequencing data generated in this study have been deposited in the NCBI Gene Expression Omnibus (GEO) database under accession code GSE168637 [https://www.ncbi.nlm.nih.gov/geo/query/acc.cgi?acc=GSE168637]. Encode datasets were downloaded with the accession numbers: H3K27me3 ENCSR059MBO [https://www.encodeproject.org/experiments/ENCSR059MBO/], H3K27me3 input ENCSR326ULS [https://www.encodeproject.org/experiments/ENCSR326ULS/], H3K4me3 ENCSR000CGO [https://www.encodeproject.org/experiments/ENCSR000CGO/] and H3K4me3 input ENCSR095IPH [https://www.encodeproject.org/experiments/ENCSR095IPH/] [https://www.encodeproject.org/experiments/ENCSR095IPH/]. The other external datasets were downloaded from NCBI Gene Expression Omnibus (GEO) [http://www.ncbi.nlm.nih.gov/geo/], with the accession numbers: Aebp2 ChIP-seq GSE83082 [https://www.ncbi.nlm.nih.gov/geo/query/acc.cgi?acc=GSE83082], Paired-

Tag GSE152020 [https://www.ncbi.nlm.nih.gov/geo/query/acc.cgi?acc=GSE152020], scATAC-seq GSE100033 [https://www.ncbi.nlm.nih.gov/geo/query/acc.cgi?acc=GSE100033], Smart-seq2 GSE151334 [https://www.ncbi.nlm.nih.gov/geo/query/acc.cgi?acc=GSE151334]. 10x scRNA-seq datasets were downloaded from the 10x Genomics website [https://www.10xgenomics.com/]. Source data are provided with this paper.

## Code availability

Custom scripts used in this study are available from [https://github.com/Fanglab-zju/scSET-seq].

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

## Acknowledgements

We thank Dr. Roel Nusse for sharing the method of Wnt3a-coated beads construction. We thank Dr. Xudong Wu for sharing the mES cell lines. We thank Hangjun Wu in the Center of Cryo-Electron Microscopy (CCEM), Zhejiang University for his technical assistance on the computing node setup. We thank the Core facility at Life Sciences Institute, Zhejiang University for the assistance on the Confocal Laser Scanning Microscopy. We thank Yibo Wang for her assistance in the analysis of immunofluorescence data. We thank Dr. Li Shen and Dr. Jiali Yu for the assistance in picking single cell. We thank Dr. Bing Ren for sharing the raw data and computer codes from Paired-Tag. This work was supported by the National Natural Science Foundation of China (Grant no. 81874153).

## Author contributions

Conceptualization, Z.S., Y.T., Y.Z., Y.F., and D.F.; Methodology, Z.S., Y.T., Y.Z., and Y.F.; Investigation, Z.S., Y.T., Y.Z., Y.F., J.J., W.Z., and D.F.; Writing – Original Draft, Z.S., Y.T., and D.F.; Writing – Review & Editing, Z.S., Y.T., Y.Z., Y.F., and D.F.; Funding Acquisition, D.F.; Supervision, D.F.

## Competing interests

The authors declare no competing interests.
