## [Peer Review File · Nature Communications]

Reviewers' Comments:

Reviewer #1:

Remarks to the Author:

In this manuscript, the authors develop a multi-omic sequencing method called SET-seq to jointly profile the transcriptome and epigenome (histone marks) in the same single cell. Once established, SET-seq is applied to investigate the question of whether Wnt-mediated asymmetrically-divided mouse ES cells have distinct gene expression and histone mark profiles in the daughter cells. The authors find that the daughter cells differ transcriptionally and epigenetically and that asymmetric cell divisions can be identified based on clustering the SET-seq data sets. In particular, H3K27me3 profiles showed stronger differences between asymmetric daughter cells as compared to H3K4me3 profiles, implying a connection between H3K27me3 and the outcome of cell division events. Observing that the proportion of asymmetric divisions is altered in Aebp2 knockout ES cells, the authors propose that the PRC2 co-factor Aebp2 regulates H3K27me3 to control cell-fate decisions in response to localized Wnt signals.

This is an interesting and timely study. The major strength of the manuscript is that the authors apply a novel method to a very interesting question and one that really makes good use of the new technique. We found that the study design was clever and well done. The major weaknesses of the current manuscript are the lack of detailed information provided about SET-seq (quality control and methods) and the conclusion about the role of Aebp2 is not supported.

The manuscript falls into three main sections:

1. Method development. On the whole, this section is clear and in general the data support the author's interpretations, although there are some important gaps that must be addressed.

1.1 The authors should provide additional benchmarking of SET-seq to established methods, and this applies to both the RNA-seq and to the histone methylation data sets. For example, it is clear that the RNA-seq data are very sparse compared to other single cell RNA-seq methods and it is important that the authors to examine this carefully, show the evidence clearly, and if appropriate to acknowledge that this is a current limitation of the technique. Similarly, it is difficult to assess the histone methylation data sets in terms of how many unique fragments were detected per single cell, how many peaks overlap with the bulk data set, and how this compares to similar published methods, e.g. Paired Tag; scChIP-seq etc.

1.2 The RNA-seq and histone methylation methods are based largely on previously published methods and this should be clearly acknowledged and cited. At the moment it is not.

1.3 The methods need much more detail in order to be followed. Not all buffers or reaction conditions are defined, for example. In addition, we also encourage the authors to include all method information rather than referring only to other studies.

2. Application of SET-seq to study Wnt-mediated asymmetric cell states. The experiments were well designed and made good use of the SET-seq method to address an important question. Specific comments:

2.1 Page 10, the authors write "To avoid the biased selection of mESCs, we picked the cells without predetermining the Nanog signal." This is fine but did the authors record the levels of Nanog signal for each cell? If so, these data should be used retroactively to confirm whether the Nanog-high / low cells have been accurately categorized by the SET-seq data.

2.3 Page 11, the authors mention clusters 0, 1 and 2, and then just underneath mention clusters 1, 2 and 3. Are these the same clusters?

2.4 The authors apply the names "naïve" and "primed" to their different cell categories. But naïve and primed have specific meanings in stem cell biology and applying these names in this study in a different way could lead to confusion.

2.5 Based on pseudotime data, the authors seem to imply that the cells in the 'mix' category are in-between naïve and primed cell categories. This conclusion is not sufficiently supported. Also, it does not seem to fit with the finding that the 'mix' cells are enriched for terms associated with neural cells.

3. Aebp2 regulates H3K27me3 to control cell-fate decisions. This final section is very weak compared to the other parts of the manuscript and it detracts from the overall quality of the paper. The model is too simplistic and the conclusion is not supported. Claims that K27me3 / Aebp2 might have causative roles in biasing symmetric/asymmetric cell divisions should be removed from the paper. Additional comments:

3.1 If Aebp2 regulates H3K27me3 to control the proportion of symmetric/asymmetric cell divisions, then why are similar effects not seen with the Ezh2 knockout cells?

3.2 Do the authors have an explanation for how Aebp2 might be connected with altered cell fate decisions and why the loss of Aebp2 might promote an increase from ~60% to ~70% in the proportion of asymmetric divisions?

3.3 Are H3K27me3 profiles changed in the Aebp2 knock out cells?

Reviewer #2:

Remarks to the Author:

This manuscript developed a new SET-seq (same cell epigenome and transcriptome sequencing) method for profiling transcriptome and epigenome in single cells. They then used this method to analyze transcriptome and epigenome (two antagonistic histone modifications: H3K4me3 and H3K27me3) in mouse embryonic stem cells induced with Wnt beads. Therefore, there are two major parts with their corresponding novelty: from the technical perspective, the authors did a rigorous condition testing of this SET-seq method and cross-comparison their results with those from other techniques. Overall, this part of the data is well presented, with a few places that require further clarification or discussion (see below #1-2 points). From the biological perspective, the authors went ahead and applied this technique on a Wnt3a coated bead-induced asymmetric cell division model. This approach is very suitable for this model to analyze molecular properties at the single-cell resolution. Their main conclusions are H4K4me3 landscape is unchanged but H3K27me3 changes in the two distinct daughter cells, and such a change correlates with the gene expression change.

Overall, this manuscript is very strong, which both uses an innovative, well described technique and makes some interesting biological findings. However, there are a few places that either need more clarification/discussion, or need more experimental data to strengthen the conclusions.

I would highly support publication of this work after these revisions are done.

1. Fig. 1d, the clustering of the three groups start to become less distinct using 100 cells. Some discussion should be included here to comment on the optimal cell number for this method.

2. Fig.1g, the authors mentioned that "The fraction of reads in peaks (FRiP) was around 2-fold greater in scSET-seq than scATAC-seq (Fig. 1g)." But no scATAC-seq data were included in this panel for comparison. Do the authors mean Fig. 1h?

3. For picking daughter cells derived from asymmetric division, the authors described the procedure as "we manually examined cells contacting to beads, and picked the proximal and distal daughter cells into 96-well plates to perform indexed scSET-seq". Due to the high mobility of culture cells in the dish, the cells may happen to be close to each other even with low cell density. On the other hand, the two daughter cells could fall apart from each other post division. How do the authors determine these two cells are from one division? Do they follow cell division using live cell imaging or do they have other cellular marker, such as the midbody, to be sure that the two picked cells indeed derive from one asymmetric cell division?

4. Fig. 2: Using Wnt3a to induce asymmetric division, there are still symmetrically dividing cells. Are they paired cells within the Mix clusters? If so, is this ratio different from the ratio estimated using the Nanog marker? To what extent do the authors think that the annotated clusters match

up with cell division modes? Some explanations here would be helpful.

5. Fig. 6: Regarding the knockout assay of *Aebp2* and *Ezh2*: (1) Do these knockouts affect the cell fate and/or differentiation of mouse embryonic stem cells? If so, how to make sure that the readouts are not due to overall change of stemness or differentiation potential of the knockout cells? (2) Also, results in Figure 6 seem to be inconsistent with the data shown in Figure 3-5 that *K27me3* changes are tightly coupled to gene expression changes. Here, knocking out *Ezh2* removes *H3K27me3* but no effect on cell fate during division? This part is confusing as one main conclusion is that *H3K27me3* plays a leading role in the changing gene expression and cell fate. (3) Since there is no change for the total level of *H3K27me3* after knocking out *AEBP2*, how about the distribution of *H3K27me3* in these cells? It would also be ideal if SET-seq technique can be used to profile gene expression changes in the mutant conditions. This would both help address the above concern and help shed light on how *Aebp2* mechanistically contributes to the mixed cell population fate.

Minor comments:

1. The references in the Introduction part for asymmetric cell division are old (more than 10 years), I suggest the authors could use more recent and updated literature.

2. Ref. 13 is incomplete.

3. In the Introduction, the "several low-input epigenomic profiling methods" should include a few others, such as the ACT-seq, ChIC-seq, etc.

4. Pg 16, "If cells in one cluster defined by epigenomic signal were random distributed, they would recapitulate the ration of clusters defined by the gene expression.' Should be ratio.

5. For the Discussion, the authors could discuss more on the biological significance of their findings. For example, what could make the difference between *H3K27me3* and *H3K4me3* during stem cell differentiation? Are these differences related to their distinct target genes or distinct modes of generating and/or maintaining these modifications? What is the significance for *Aebp2* to maintain the "mixed" population? Does this mixed cluster of cells, which is the majority of cells, represent an intermediate differentiation state? If so, what is the biological significance for having such a state?

Reviewer #3:

Remarks to the Author:

Sun et al. present a method with an ability for single-cell concurrent measurement of transcriptome and histone modifications. Through several optimizations, the authors pushed it from low input samples to single cells by profiling RNA/*H3K4me3* and RNA/*H3K27me3* with mESCs treated with localized *Wnt3a* to study the mechanism of asymmetric cell division. In this work, they found that *H3K27me3*, not *H3K4me3*, is correlated with the change in gene expression during asymmetric cell division. Further, they claimed that *Aebp2* as a subunit of PRC2 complex was involved in this event by CRISPR/Cas9 technique in mESCs.

Overall, scSET-seq represent a low throughput technique by mouth pipetting single cells into 96-well plates. Particularly, a similar method, Paired-tag, outperforms scSET-seq in many aspects. Although RNA parts in scSET-seq display comparable detected gene numbers to Smart-seq2, the authors did not a good job to fully benchmark or give necessary information on genomic fragments on ChIP parts per cell. Further, how many independent experiments for scSET-seq on *H3K4me* and *H3K27me3* were performed is also absent throughout the manuscript. Given lack of essential information, it is hard to appreciate both the data quality and the biological finding. In addition, I am not convinced by the specific role of *Aebp2* only based on immunostaining. The marginal difference appeared barely significant to me.

Major concerns:

1. The authors conducted several analyses in Fig. 1 to assess the data quality of SET-seq.

However, the global evaluation of SET-seq data quality in single cells lacks. The authors never quantified the histone modification partition. The QC information including mapping rate, peak numbers, peaks overlapped with those from conventional bulk ChIP, total UMIs, genes detected, unique reads detected in single cells needs to be clearly presented on.

2. To evaluate the signal-to-noise ratio of SET-seq, the authors plotted ROC and visualize *H3K27me3* signal in Supplemental Fig. S5c and Fig. 1g. It is not clear

3. In Fig. 1d, three genes clusters were identified based on *H3K4me3*/*H3K27me3* signal and RNA level. What are the genes in each cluster? Showing several typical genes or performing GO term

analysis for H3K4me3-high, H3K27me3-high, bivalent clusters would strengthen the clarification of data quality.

4. As shown in Fig. 2, the "Mix" cell population was identified as an intermediate between "Naïve" and "Prime" state. Mix cluster was enriched for differentiation -related term such as neuron interaction (Figure S8). Why differentiation terms were not observed in the "Prime" population? Furthermore, Nanog and Rex1 broadly expressed in Mix cluster, which is not seen in Naïve cluster (Fig. S9). This is confusing.

5. In Fig. 2j, two trajectories were identified by monocle2. The authors should elaborate on the lineage relationship here, which is also an additional clarification of the "Mix" cells (mainly occur in one branch) as I mentioned above.

6. Most of the Mix/Naïve/Prime cluster markers shown in Fig. 3a are not specific for each cluster both on the RNA and histone level.

7. Using module analysis to calculate the correlation between transcriptome and histone marks, the authors found that H3K27me3 was more correlated with gene expression. This is inconsistent with many previously reported studies, i.e. H3K4me3 is an active mark for gene expression. Given that very little is done to evaluate the single-cell data quality, I am rather concerned about the potential noise and data sparsity beyond the biological conclusions.

Minor points:

1. The language demands sufficient work to improve it as the current format more or less prevents one from fully appreciating this work.

2. Many typos exist in the manuscript. For example, "stem cell-like statues" should be "stem cell-like status"; "Figure ledged" should be "Figure legend"; "virtualize" should be "visualize".

We thank the reviewers for their time and energy to help us improve this manuscript. We included the detailed methods and compared SET-seq with other published methods including Smart-Seq2, Paired-Tag and 10x genomics. Additional data quality controls were done for the data quality and the results showed that scSET-seq was similar to Smart-Seq2 and Paired-Tag on the performances of gene expression and histone marks profiling. We further profiled gene expressions and H3K27me3 in Aebp2 KO cells to address the concerns about the functions of Aebp2 and Ezh2. Further investigations of Mix cluster cells were performed as the reviewer requested and we found that these cells were branched to a differentiation trajectory. Daughter cells paired inside the Mix cluster had a similar ratio to the cells symmetrically divided as detected by Nanog signal. Please see the detailed response to every point below.

Reviewer #1 (Remarks to the Author):

In this manuscript, the authors develop a multi-omic sequencing method called SET-seq to jointly profile the transcriptome and epigenome (histone marks) in the same single cell. Once established, SET-seq is applied to investigate the question of whether Wnt-mediated asymmetrically-divided mouse ES cells have distinct gene expression and histone mark profiles in the daughter cells. The authors find that the daughter cells differ transcriptionally and epigenetically and that asymmetric cell divisions can be identified based on clustering the SET-seq data sets. In particular, H3K27me3 profiles showed stronger differences between asymmetric daughter cells as compared to H3K4me3 profiles, implying a connection between H3K27me3 and the outcome of cell division events. Observing that the proportion of asymmetric divisions is altered in Aebp2 knockout ES cells, the authors propose that the PRC2 co-factor Aebp2 regulates H3K27me3 to control cell-fate decisions in response to localized Wnt signals.

This is an interesting and timely study. The major strength of the manuscript is that the authors apply a novel method to a very interesting question and one that really makes good use of the new technique. We found that the study design was clever and well done. The major weaknesses of the current manuscript are the lack of detailed information provided about SET-seq (quality control and methods) and the conclusion about the role of Aebp2 is not supported.

The manuscript falls into three main sections:

1. Method development. On the whole, this section is clear and in general the data support the author's interpretations, although there are some important gaps that must be addressed.

1.1 The authors should provide additional benchmarking of SET-seq to established methods, and this applies to both the RNA-seq and to the histone methylation data sets. For example, it is clear that the RNA-seq data are very sparse compared to other single cell RNA-seq methods and it is important that the authors to examine this carefully, show the evidence clearly, and if appropriate to acknowledge that this is a current limitation of the technique. Similarly, it is difficult to assess the histone methylation data sets in terms of how many unique fragments were detected per single cell, how many peaks overlap with the bulk data set, and how this compares to similar published methods, e.g. Paired Tag; scChIP-seq etc.

We thank the reviewer to help us improve the quality control for scSET-seq. And we also thank Dr. Bing Ren for sharing with us the raw data of Paired-tag and 10x genomics to draw a comparison panel. We first performed quality controls for the gene expression profiles. The

library construction method was similar between scSET-seq and Smart-Seq2 which were based on the whole cDNA fragments, and between 10x genomics and Paired-Tag which were mainly detecting 3' end of cDNA. As previously reported¹, the ratios of intragenic reads and intronic reads were both lower in Smart-Seq2 when compared with 10x genomics (Supplemental Fig. S6b and S6c). The scSET-seq and Paired-Tag were similar to Smart-Seq2 and 10x genomics based on the ratios of intragenic and intronic reads, respectively. Like SmartSeq2, the gene expression libraries of scSET-seq were constructed without Unique Molecular Identifiers (UMIs), which were widely used in the 3' RNA seq, like 10x genomics, to distinguish unexpected PCR duplications and rare mutation variants. Page 9, last paragraph. We also discussed other potential improvements in the discussion section. Page 26, last paragraph.

As the reviewer suggested, we performed other analyses for the histone methylation profiles. Please note, for a direct comparison with Paired-Tag, we calculated the FRiP by the same method in Paired-Tag paper (We previously used the peaks from bulk-seq as the positive reference peaks. For the Paired-Tag, peaks called from merged single cell data were set as the positive reference peaks for this new comparison). The FRiPs from scSET-seq slightly changed compared to our old method. Nevertheless, the FRiPs were comparable and similar between scSET-seq and Paired-Tag, both of which were higher than scATAC-seq (Fig. 1h). We also noticed that the FRiPs were higher for H3K4me3 than that for H3K27me3 in both Paired-Tag and scSET-seq. The mean unique fragments detected in H3K27me3 and H3K4me3 scSET-seq were around 70 and 200 times more than those of Paired-Tag, respectively (Supplemental Fig. S6d). This was simply because we sequenced more reads in the scSET-seq. Moreover, we called peaks from the bulk SET-seq of 10,000 cells, merged scSET-seqs, and ENCODE data for H3K27me3 (ENCSR059MBO) and H3K4me3 (ENCSR000CGO), respectively. The peaks called from bulk SET-seq were largely overlapped with corresponding ENCODE data. Peaks called from merged scSET-seqs were overlapped well with bulk SET-seq and ENCODE data (Supplemental Fig. S6e and S6f).

To further provide the quality control results and basic information for the sequencing results, we include a metadata file for H3K4me3 and H3K27me3 scSET-seq in Figure 1, including total reads, mapped reads, uniquely mapped reads, unique fragments, and FRiP (Supplemental Table S1).

We include a metadata file for scSET-seq with Wnt3a beads and included the information about total reads, mapped reads, uniquely mapped reads, unique fragments, detected gene number, FRiP, Cluster information of UMAP_1 and UMAP_2, and cell barcodes (Supplemental Table S3).

We include a table presenting the total reads and uniquely mapped reads for all the bulk sequencing (Supplemental Table S7).

1.2 The RNA-seq and histone methylation methods are based largely on previously published methods and this should be clearly acknowledged and cited. At the moment it is not.

We thank the reviewer for reminding us to cite the methods. We have modified the manuscript to cite the previous methods for epigenome and transcriptome in the results, discussion and method sections. Result section: Page 5, last paragraph. Discussion section: Page 21, last paragraph. Method section: Page 31, first paragraph.

1.3 The methods need much more detail in order to be followed. Not all buffers or reaction conditions are defined, for example. In addition, we also encourage the authors to include all method information rather than referring only to other studies.

We thank the reviewer for this very helpful suggestion to describe the method. We have rewritten the method section to include detailed receipt of buffers, reaction conditions, procedures, and resources of reagent.

2. Application of SET-seq to study Wnt-mediated asymmetric cell states. The experiments were well designed and made good use of the SET-seq method to address an important question. Specific comments:

2.1 Page 10, the authors write “To avoid the biased selection of mESCs, we picked the cells without predetermining the Nanog signal.” This is fine but did the authors record the levels of Nanog signal for each cell? If so, these data should be used retroactively to confirm whether the Nanog-high / low cells have been accurately categorized by the SET-seq data.

We monitored the dividing of parental cells with white light to confirm that two daughter cells were from the same parental cell. The intensities of Nanog were not recorded since we needed to proceed the cells within a short time. In addition, cells migrated fast and lots of two paired daughter cells moved to a slightly different distance in z axis, leading to the loss of focus in one of the cells when immunofluorescence images were taken. It's too difficult to collect enough cells for the scSET-seq if all cells were needed to be recorded with the immunofluorescence intensities. However, as the reviewer pointed out, it would be interesting to record this information for the comparison of cell clusters and immunofluorescence intensities in the further experiment. We added this to the discussion section to point out this. Page 26, last paragraph.

2.3 Page 11, the authors mention clusters 0, 1 and 2, and then just underneath mention clusters 1, 2 and 3. Are these the same clusters?

We thank the reviewer for this correction. They are the same clusters. Cluster 0 was mislabeled as Cluster 3 underneath the description. We corrected the Cluster 3 to Cluster 0 accordingly.

2.4 The authors apply the names “naïve” and “primed” to their different cell categories. But naïve and primed have specific meanings in stem cell biology and applying these names in this study in a different way could lead to confusion.

We thank the reviewer for this useful suggestion. These names may lead to a confusing understanding. We renamed the “naïve” and “primed” clusters to “Proxi” and “Distal” clusters, respectively. All the names in the revised manuscript were changed to the new names accordingly.

For the following response to reviewers' comments, we keep the “naïve” and “primed” names to be consistent to the questions in this response letter.

2.5 Based on pseudotime data, the authors seem to imply that the cells in the ‘mix’ category are in-between naïve and primed cell categories. This conclusion is not sufficiently supported. Also, it does not seem to fit with the finding that the ‘mix’ cells are enriched for terms associated with neural cells.

We thank the reviewer to point out this misleading interpretation. The pseudotime analysis showed a progression of Naïve cluster at the beginning of the trajectory to Prime cluster was at the end. The Mix cluster separating from the pseudotime analysis was detected in one branch alongside the trajectory. This trajectory indicated a differentiation progress from the early Naïve cluster to the late Prime cluster, whereas the Mix cluster occurred as a distinct differentiation lineage to differ from the asymmetrically divided cells. In addition, as the reviewer 3 promoted us to look into GO terms more carefully. We found Prime cluster was

also enriched for several nucleoside triphosphate metabolic process terms, which participated in erythroid differentiation², T cell lineage differentiation³, and neurogenesis⁴. So, the missing terms of differentiation might be overridden by the nucleoside triphosphate metabolic process terms in Prime cluster. We modified the description of Mix cluster in the result section to avoid the misleading interpretation. Page 14, last paragraph.

3. Aebp2 regulates H3K27me3 to control cell-fate decisions. This final section is very weak compared to the other parts of the manuscript and it detracts from the overall quality of the paper. The model is too simplistic and the conclusion is not supported. Claims that K27me3 / Aebp2 might have causative roles in biasing symmetric/asymmetric cell divisions should be removed from the paper. Additional comments:

We acknowledge that the causative role of Aebp2 in the Wnt induced asymmetric cell division was not fully supported. We removed the statement of causative relation in the revised manuscript, performed other analyses as shown below, and soften the language to only state that knockout of Aebp2 increased the ratio of daughter cells asymmetrically expressing Nanog.

3.1 If Aebp2 regulates H3K27me3 to control the proportion of symmetric/asymmetric cell divisions, then why are similar effects not seen with the Ezh2 knockout cells?

Through the scSET-seq, we found the components of PRC2 were changed differently among the cell clusters (Fig. 6a), indicating that these marker genes were controlled by different enrichments of H3K27me3. The locus specific gain or loss of H3K27me3 was important for the asymmetric cell division. The knockout of Ezh2 largely depleted H3K27me3 in mESCs which significantly changed the epigenome. So that the cell division pattern was different from the locus-specific change caused by Aebp2 KO. In addition, with the H3K27me3 profiling suggest by the reviewer, we found H3K27me3 increased when Aebp2 was knocked out, especially at the Mix marker genes. This difference of subsequently changed H3K27me3 may also lead to distinct cell behavior between Ezh2 and Aebp2 KO during Wnt3a induced asymmetric cell division.

3.2 Do the authors have an explanation for how Aebp2 might be connected with altered cell fate decisions and why the loss of Aebp2 might promote an increase from ~60% to ~70% in the proportion of asymmetric divisions?

We thank the reviewer for this comment. In line with the reviewer's suggestion in comment 3.3, we profiled H3K27me3 in Aebp2 KO cells. Two biological repeats were conducted for each sequencing in the two KO cell lines. These two repeats were correlated well (Supplemental Table S8). Moreover, the H3K27me3 sequencing results were correlated well between two KO cell lines (Supplemental Fig. S14f). We then merged the two cell lines and two repeats to detect the consistent changes of H3K27me3. We called 43,747 and 59,770 H3K27me3 peaks in WT and Aebp2 KO cells respectively. While 30,332 peaks overlapped between WT and Aebp2 KO cells, 29,438 peaks were unique to Aebp2 KO cells (Supplemental Fig. S14g). We then compared the enrichments of Aebp2 and H3K27me3 at cluster marker genes. In wild type cells, the enrichments of Aebp2 and H3K27me3 were similar among three cluster, whereas the Naïve and Prime cluster markers showed higher enrichments than Mix cluster marker genes. Only H3K27me3 at the Mix marker genes were increased when Aebp2 was knocked out (Fig 6e and 6f). These results were consistent with previous report that H3K27me3 showed a minor increase at Aebp2 target genes when Aebp2 was knocked out^{5,6}. It's possible that the increased H3K27me3 at the Mix marker genes was less in Aebp2 KO cells, leading to the failure of the activation of marker genes during the

Wnt induced asymmetric cell division and subsequently the reduction, but not totally abolishment, of the symmetrically divided cells. These data were included in Page 20 – 21.

3.3 Are H3K27me3 profiles changed in the Aebp2 knock out cells?

We profiled H3K27me3 in two of the Aebp2 KO cell lines with two biological repeats for each cell line. The results of two biological repeats and two KO cell lines were correlated well (Supplemental Table S8 and Supplemental Fig. S14f). We called 43,747 and 59,770 H3K27me3 peaks in WT and Aebp2 KO cells respectively. While 30,332 peaks overlapped between WT and Aebp2 KO cells, 29,438 peaks were unique to Aebp2 KO cells (Supplemental Fig. S14g). We further analyzed the enrichment of H3K27me3 at peaks with or without overlapped Aebp2 peaks that were detected in ENCODE Aebp2 ChIP-seq dataset (GSE83082). H3K27me3 was higher at the H3K27me3 peaks that were overlapped with Aebp2 peaks when compared with those not overlapped (Supplemental Fig. S14h). More importantly, in Aebp2 KO cells, H3K27me3 increased at the peaks overlapped with Aebp2 peaks, but not changed at the peaks without Aebp2 peaks. This was consistent with previous reports that, although Aebp2 increased the enzymatic activity of PRC2 *in vitro*, cells without Aebp2 exhibited elevated H3K27me3 at Aebp2 targeting sites^{5, 6}. It's possibly due to an increased presence of other PRC2 subcomplexes or formation of hybrid PRC2 subcomplexes at the Aebp2 depleted loci, leading to elevated PRC2 recruitment to target loci⁷. These data were included in Page 20 – 21.

Reviewer #2 (Remarks to the Author):

This manuscript developed a new SET-seq (same cell epigenome and transcriptome sequencing) method for profiling transcriptome and epigenome in single cells. They then used this method to analyze transcriptome and epigenome (two antagonistic histone modifications: H3K4me3 and H3K27me3) in mouse embryonic stem cells induced with Wnt beads. Therefore, there are two major parts with their corresponding novelty: from the technical perspective, the authors did a rigorous condition testing of this SET-seq method and cross-comparison their results with those from other techniques. Overall, this part of the data is well presented, with a few places that require further clarification or discussion (see below #1-2 points). From the biological perspective, the authors went ahead and applied this technique on a Wnt3a coated bead-induced asymmetric cell division model. This approach is very suitable for this model to analyze molecular properties at the single-cell resolution. Their main conclusions are H3K4me3 landscape is unchanged but H3K27me3 changes in the two distinct daughter cells, and such a change correlates with the gene expression change. Overall, this manuscript is very strong, which both uses an innovative, well described technique and makes some interesting biological findings. However, there are a few places that either need more clarification/discussion, or need more experimental data to strengthen the conclusions. I would highly support publication of this work after these revisions are done.

1. Fig. 1d, the clustering of the three groups start to become less distinct using 100 cells. Some discussion should be included here to comment on the optimal cell number for this method.

We thank the reviewer for this very helpful suggestion. Because two histone marks, H3K4me3 and H3K27me3, alongside gene expressions needed to be detected at the same gene, less genes were detected when 100 cells were used as the starting material. The classified clusters from 100 cells were less distinct comparing to the clusters from higher amounts of cells. This data indicated that, when bulk cells were used to detect both H3K4me3 and H3K27me3 along with gene expression signals, around 1,000 cells were optimal for this

specific application. These interpretations were included in the corresponding results section. Page 9, first paragraph.

2. Fig. 1g, the authors mentioned that “The fraction of reads in peaks (FRiP) was around 2-fold greater in scSET-seq than scATAC-seq (Fig. 1g).” But no scATAC-seq data were included in this panel for comparison. Do the authors mean Fig. 1h?

We thank the reviewer for this correction. The reference was mislabeled. We have corrected the citation of Fig. 1g and Fig. 1h. Page 10, last paragraph.

3. For picking daughter cells derived from asymmetric division, the authors described the procedure as “we manually examined cells contacting to beads, and picked the proximal and distal daughter cells into 96-well plates to perform indexed scSET-seq”. Due to the high mobility of culture cells in the dish, the cells may happen to be close to each other even with low cell density. On the other hand, the two daughter cells could fall apart from each other post division. How do the authors determine these two cells are from one division? Do they follow cell division using live cell imaging or do they have other cellular marker, such as the midbody, to be sure that the two picked cells indeed derive from one asymmetric cell division?

We thank the reviewer for this suggestion to specify how cells were collected. As the reviewer pointed out, the behaviors of cells were monitored by live cell imaging system (DeltaVision Elite Deconvolution Microscope). The dividing cells were traced with white light to follow the cell mobility. Two cells divided from the same parental cell were proceeded. We included these descriptions in the method section. Page 33, first paragraph.

4. Fig. 2: Using Wnt3a to induce asymmetric division, there are still symmetrically dividing cells. Are they paired cells within the Mix clusters? If so, is this ratio different from the ratio estimated using the Nanog marker? To what extent do the authors think that the annotated clusters match up with cell division modes? Some explanations here would be helpful.

The ratio of cells in Mix cluster was 53.4% of all cells and the ratio of paired cells within the Mix cluster was 56.7% of cells in Mix cluster. So, the ratio of paired cells inside the Mix clusters was 30.3% of total cells. This is similar to previous reports showing that the symmetrically dividing cells were around 25~30%. The reversely divided cells, as determined by the distribution of paired cells with proximal cell in Prime cluster and distal cell in Naïve cluster, were around 10% of total cells. This was also similar to previous observations that around 15% of cells were reversely divided when the cell fates were determined by the Nanog signals. In addition, the asymmetrically divided cells composed around 60% of total cells which was similar to the previously reported ratio (~60%). The annotated clusters reflected, at the level of ratios of cell behaviors, the previous reported cell division modes. We added these interpretations in the corresponding result section. Page 14, first paragraph.

5. Fig. 6: Regarding the knockout assay of Aebp2 and Ezh2: (1) Do these knockouts affect the cell fate and/or differentiation of mouse embryonic stem cells? If so, how to make sure that the readouts are not due to overall change of stemness or differentiation potential of the knockout cells? (2) Also, results in Figure 6 seem to be inconsistent with the data shown in Figure 3-5 that K27me3 changes are tightly coupled to gene expression changes. Here, knocking out Ezh2 removes H3K27me3 but no effect on cell fate during division? This part is confusing as one main conclusion is that H3K27me3 plays a leading role in the changing gene expression and cell fate. (3) Since there is no change for the total level of H3K27me3 after knocking out AEBP2, how about the distribution of H3K27me3 in these cells? It would

also be ideal if SET-seq technique can be used to profile gene expression changes in the mutant conditions. This would both help address the above concern and help shed light on how Aebp2 mechanistically contributes to the mixed cell population fate.

We thank the reviewer to help us improve the analysis of Aebp2 and Ezh2.

We performed SET-seq in Aebp2 knockout cell for H3K27me3 and gene expression. Two independent experiments were done for each KO cell line. These two repeats of gene expression and H3K27me3 were correlated well respectively (Supplemental Table S6 and S8). In addition, the two Aebp2 KO clones showed a consistent gene expression profile which were differ from the wild-type cell (Supplemental Fig. S14c). We then merged the gene expression profiles of two Aebp2 KO clones to get the consistently changed genes, which were defined with a $\log_2(\text{foldchange}) > 1$ and p value less than 0.05 (Supplemental Fig. S14d). 1414 genes were up-regulated and 1403 genes were down-regulated. In the total of 321 marker genes, only 33 and 47 were up- and down-regulated in Aebp2 KO cells respectively (Supplemental Fig. S14d and S14e), suggesting that the expressions of cluster marker genes were not largely altered. The pluripotency marker genes, including Oct4, Nanog, and Sox2 were not changed when Aebp2 was knocked out. In addition, previous study had shown that Nanog and Oct4 was not changed when Ezh2 was knocked out in mESCs⁸. As the reviewer pointed out, Aebp2 and Ezh2 knockout could affect the differentiation of mESCs⁷. These readouts of cell differentiation may reflect the differentiation potential of knockout cells. We think this agree with our findings that, in the Wnt3a induced asymmetric cell division, the differentiation potential of knockout cells was repressed to reduce the symmetric cell division.

The knockout of Ezh2 largely abolished H3K27me3 in mESCs which significantly changed the epigenome. The components of PRC2 were changed differently among the cell clusters indicating that these marker genes were controlled by different enrichments of H3K27me3. When the overall levels of H3K27me3 were abolished, other epigenomic information carrier, like H2AK119ub and Ezh1, can function with a complementary effect on the Ezh2 dependent H3K27me3^{8,9}. So, not the total level but the locus specific gain or loss of H3K27me3 was important for the asymmetric cell division.

As stated above, we also analyzed the distributions of H3K27me3 upon Aebp2 KO.

We called 43,747 and 59,770 H3K27me3 peaks in WT and Aebp2 KO cells respectively. While 30,332 peaks overlapped between WT and Aebp2 KO cells, 29,438 peaks were unique to Aebp2 KO cells (Supplemental Fig. S14g). We further analyzed the enrichment of H3K27me3 at peaks with or without overlapped Aebp2 peaks that were detected in ENCODE Aebp2 ChIP-seq dataset (GSE83082). H3K27me3 was higher at the H3K27me3 peaks that were overlapped with Aebp2 peaks when compared with those not overlapped (Supplemental Fig. S14h). More importantly, in Aebp2 KO cells, H3K27me3 increased at the peaks overlapped with Aebp2 peaks, but not changed at the peaks without Aebp2 peaks. This was consistent with previous reports that, although Aebp2 increased the enzymatic activity of PRC2 *in vitro*, cells without Aebp2 exhibited elevated H3K27me3 at Aebp2 targeting sites^{5,6}. It's possibly due to an increased presence of other PRC2 subcomplexes or formation of hybrid PRC2 subcomplexes at the Aebp2 depleted loci, leading to elevated PRC2 recruitment to target loci⁷. We then compared the enrichments of Aebp2 and H3K27me3 at cluster marker genes that were identified by gene expression. Compare to Mix cluster marker genes, Aebp2 was high at Proxi and Dista cluster marker genes in wild-type cells (Fig. 6e). Like Aebp2, H3K27me3 was higher at marker genes of Proxi and Dista cluster than those of Mix cluster. In addition, H3K27me3 increased at Mix cluster marker genes and slightly decreased at Proxi cluster marker genes, when Aebp2 was knocked out (Fig. 6f). The compensation of

increased H3K27me3 may lead to the observation that not all daughter cells asymmetrically expressed Nanog when Aebp2 was knocked out.

These data were included in Page 20 – 21.

Minor comments:

1. The references in the Introduction part for asymmetric cell division are old (more than 10 years), I suggest the authors could use more recent and updated literature.

We thank the reviewer for this very helpful suggestion. We recited the review papers with recent literatures.

2. Ref. 13 is incomplete.

We thank the reviewer for this carefully reading of the manuscript. We corrected this citation.

3. In the Introduction, the “several low-input epigenomic profiling methods” should include a few others, such as the ACT-seq, ChIC-seq, etc.

We included these suggested methods and others including scChIP-Seq, scChIC-seq, TAF-ChIP, muChIP-seq, LIFE-ChIP-seq, TCL, and SurfaceChIP-seq.

4. Pg 16, “If cells in one cluster defined by epigenomic signal were random distributed, they would recapitulate the ration of clusters defined by the gene expression.” Should be ratio.

We thank the reviewer for helping us improve the language. We corrected this typo and improved the writing with other corrections.

5. For the Discussion, the authors could discuss more on the biological significance of their findings. For example, what could make the difference between H3K27me3 and H3K4me3 during stem cell differentiation? Are these differences related to their distinct target genes or distinct modes of generating and/or maintaining these modifications? What is the significance for Aebp2 to maintain the "mixed" population? Does this mixed cluster of cells, which is the majority of cells, represent an intermediate differentiation state? If so, what is the biological significance for having such a state?

We thank the reviewer for this very helpful suggestion. We discussed more about the significance of the epigenetic alternation and cell differentiation as the reviewer suggested.

During the progression of differentiation, stem cells become restricted in their differentiation potential. The dynamic changes in histone modifications and associated chromatin structure are proposed to stabilize lineage-specific gene expression and regulate the differentiation potential. Remarkably, the differentiation process could be reprogrammed, by manipulating with small molecule inhibitors or transcriptional factors, to turn cells into stem cell-like state or to different cell lineages. The chromatin assembly factor-1 (CAF-1), a histone chaperone responsible for the deposition of histone H3.1 to maintain chromatin structure, is discovered to play as a potent barrier in the reversion of pluripotent cells to a totipotent-like state, further implicating the essential of chromatin structure in the regulation of cellular plasticity. H3K27me3, which is usually linked to repressed gene expression, plays important roles in mammalian embryonic development and induced pluripotent stem cell (iPSC) generation. Generally, H3K4me3 is associated with active gene expression and can also regulate cell fate decision. In mESCs, H3K27me3 and H3K4me3 enrich at same promoters as a bivalent histone mark which primes associated genes for the rapid activation during development. Studies have shown that epigenomic profiles in hESCs and primary fibroblasts are drastically different, whereas the most changes arise from the repressive histone marks, including

H3K27me3. Introduction of a heterozygous Y641F mutation in Ezh2, which modifies the ratio of H3K27me2 and H3K27me3 in mESCs, is sufficient for the gain and suppression of cell-lineage specific gene expression and cellular phenotypes. Moreover, the global loss of H3K27me3 marks facilitates the generation of iPSCs in mice and humans. Like H3K27me3, repression of H3K4me3 can improve the efficiency and blastocyst quality in the somatic cell nuclear transfer. Knockout of Mll2, which is responsible for the methylation of H3K4me3, results in impaired embryoid body formation, failing to activate or exhibiting delayed activation kinetics of many bivalent genes that are key regulators in embryonic development and differentiation. Mechanistically, MLL2 methylates H3K4 on many bivalent genes and protects these genes from repression via repelling PRC2 and DNA methylation machineries. As previously reported, we also see a minor increase of H3K27me3 at promoters when Aebp2 is knocked out. AEBP2 is important for the distinction of the PRC2 subcomplexes. It's possibly due to an increased presence of other PRC2 subcomplexes or formation of hybrid PRC2 subcomplexes, leading to elevated PRC2 recruitment to target genes. Knockout of Aebp2, but not Ezh2, increases the ratio of mESCs expressing Nanog asymmetrically with localized Wnt signal. The delicate and localized programming of H3K27me3, as determined by Aebp2, is needed during the asymmetric cell division. It would be interesting to investigate how the changes of epigenome information at subset of chromatin loci, but not the total levels, participate in the regulation of cell differentiation. The identification of Mix cluster which was branched from the main trajectory provides new insights to the asymmetric cell division, where the divided cells transferred to a different lineage other than the typical asymmetrically divided cells. Page 24-25.

Reviewer #3 (Remarks to the Author):

Sun et al. present a method with an ability for single-cell concurrent measurement of transcriptome and histone modifications. Through several optimizations, the authors pushed it from low input samples to single cells by profiling RNA/H3K4me3 and RNA/H3K27me3 with mESCs treated with localized Wnt3a to study the mechanism of asymmetric cell division. In this work, they found that H3K27me3, not H3K4me3, is correlated with the change in gene expression during asymmetric cell division. Further, they claimed that Aebp2 as a subunit of PRC2 complex was involved in this event by CRISPR/Cas9 technique in mESCs. Overall, scSET-seq represent a low throughput technique by mouth pipetting single cells into 96-well plates. Particularly, a similar method, Paired-tag, outperforms scSET-seq in many aspects. Although RNA parts in scSET-seq display comparable detected gene numbers to Smart-seq2, the authors did not a good job to fully benchmark or give necessary information on genomic fragments on ChIP parts per cell. Further, how many independent experiments for scSET-seq on H3K4me and H3K27me3 were performed is also absent throughout the manuscript. Given lack of essential information, it is hard to appreciate both the data quality and the biological finding. In addition, I am not convinced by the specific role of Aebp2 only based on immunostaining. The marginal difference appeared barely significant to me.

We performed additional quality controls for the transcriptional and epigenomic profiling results. We also compared these with other low-input based method including Smart-Seq2, scATAC-seq, Paired-Tag and 10x genomics. In general, we detected similar data quality between scSET-seq and Smart-Seq2 in gene expression and similar data quality between scSET-seq and Paired-Tag in epigenomic profiling. The throughput of scSET-seq can be increased by using FACS based cell sorting, we used mouth pipetting in the Wnt3a experiments because the divided cells cannot be selected and sorted by FACS. The

throughput of scSET-seq is the same as the Smart-Seq2 which can proceed several hundreds of cells in one experiment. We acknowledged, in the discussion section, the throughput is less than Paired-Tag because two steps FACS sorting can be used in the Paired-Tag to further increase the input cell number.

scSET-seq for gene expression was conducted through 3 independent experiments (Page 9, last paragraph). scSET-seq for H3K4me3 and H3K27me3 in Fig. 1 were performed with 5 biological independent experiments respectively (Page 10, last paragraph). scSET-seq for H3K4me3 and H3K27me3 in Wnt3a-induced asymmetric cell division were conducted from 10 and 6 independent experiments, respectively (Page 12, first paragraph). We included this information in the results section.

As the response to reviewer #1, we removed the statement of causative relation in the revised manuscript, performed additional analyses for the gene expression and H3K27me3 alternations as shown below, and soften the language to only state that knockout of Aebp2 increased the ratio of daughter cells expressing Nanog asymmetrically.

Major concerns:

1. The authors conducted several analyses in Fig. 1 to assess the data quality of SET-seq. However, the global evaluation of SET-seq data quality in single cells lacks. The authors never quantified the histone modification partition. The QC information including mapping rate, peak numbers, peaks overlapped with those from conventional bulk ChIP, total UMIs, genes detected, unique reads detected in single cells needs to be clearly presented on.

We thank the reviewer to advise us to perform more quality controls for the sequencing results. In line with the comments from other reviewers, we did quality controls to evaluate the gene expressions and histone modifications in single cells as the reviewer suggested. The basic information about each sequencing was provided in metadata files.

We copied previous response to quality control below for reading convenience:

We also thank Dr. Bing Ren for sharing with us the raw data of Paired-tag and 10x genomics to draw a comparison panel. We first performed quality controls for the gene expression profiles. The library construction method was similar between scSET-seq and Smart-Seq2 which were based on the whole cDNA fragments, and between 10x genomics and Paired-Tag which were mainly detecting 3' end of cDNA. As previously reported¹, the ratios of intragenic reads and intronic reads were both lower in Smart-Seq2 when compared with 10x genomics (Supplemental Fig. S6b and S6c). The scSET-seq and Paired-Tag were similar to Smart-Seq2 and 10x genomics based on the ratios of intragenic and intronic reads, respectively. Like SmartSeq2, the gene expression libraries of scSET-seq were constructed without Unique Molecular Identifiers (UMIs), which were widely used in the 3' RNA seq, like 10x genomics, to distinguish unexpected PCR duplications and rare mutation variants. Page 9, last paragraph. We also discussed other potential improvements in the discussion section. Page 26, last paragraph.

As the reviewer suggested, we performed other analyses for the histone methylation profiles. Please note, for a direct comparison with Paired-Tag, we calculated the FRiP by the same method in Paired-Tag paper (We previously used the peaks from bulk-seq as the positive reference peaks. For the Paired-Tag, peaks called from merged single cell data were set as the positive reference peaks for this new comparison). The FRiPs from scSET-seq slightly changed compared to our old method. Nevertheless, the FRiPs were comparable and similar between scSET-seq and Paired-Tag, both of which were higher than scATAC-seq (Fig. 1h).

We also noticed that the FRiPs were higher for H3K4me3 than that for H3K27me3 in both Paired-Tag and scSET-seq. The mean unique fragments detected in H3K27me3 and H3K4me3 scSET-seq were around 70 and 200 times more than those of Paired-Tag, respectively (Supplemental Fig. S6d). This was simply because we sequenced more reads in the scSET-seq. Moreover, we called peaks from the bulk SET-seq of 10,000 cells, merged scSET-seqs, and ENCODE data for H3K27me3 (ENCSR059MBO) and H3K4me3 (ENCSR000CGO), respectively. The peaks called from bulk SET-seq were largely overlapped with corresponding ENCODE data. Peaks called from merged scSET-seqs were overlapped well with bulk SET-seq and ENCODE data (Supplemental Fig. S6e and S6f).

To further provide the quality control results and basic information for the sequencing results, we include a metadata file for H3K4me3 and H3K27me3 scSET-seq in Figure 1, including total reads, mapped reads, uniquely mapped reads, unique fragments, and FRiP (Supplemental Table S1).

We include a metadata file for scSET-seq with Wnt3a beads and included the information about total reads, mapped reads, uniquely mapped reads, unique fragments, detected gene number, FRiP, Cluster information of UMAP_1 and UMAP_2, and cell barcodes (Supplemental Table S3).

We include a table presenting the total reads and uniquely mapped reads for all the bulk sequencing (Supplemental Table S7).

2. To evaluate the signal-to-noise ratio of SET-seq, the authors plotted ROC and visualize H3K27me3 signal in Supplemental Fig. S5c and Fig. 1g. It is not clear

We thank the reviewer for helping us present the results clearly. We enlarged the figures to show the panels clearly. We included the detailed methods for the calculation of ROC in the method section, Page 36, last paragraph. In addition, the FRiPs were similar between scSET-seq and Paired-Tag for histone marks. We also called peaks from the bulk SET-seq of 10,000 cells, merged scSET-seqs, and ENCODE data for H3K27me3 (ENCSR059MBO) and H3K4me3 (ENCSR000CGO), respectively. The peaks called from bulk SET-seq were largely overlapped with corresponding ENCODE data. Peaks called from merged scSET-seqs were overlapped well with bulk SET-seq and ENCODE data (Supplemental Fig. S6e and S6f).

3. In Fig. 1d, three genes clusters were identified based on H3K4me3/H3K27me3 signal and RNA level. What are the genes in each cluster? Showing several typical genes or performing GO term analysis for H3K4me3-high, H3K27me3-high, bivalent clusters would strengthen the clarification of data quality.

We thank the reviewer for this helpful suggestion. We performed GO analysis in the three clusters of genes as the reviewer suggested (Supplemental Fig. S5d). Genes with high H3K27me3 were mainly enriched for ion transport associated terms. Genes with high H3K4me3 were enriched in cell cycle, signal transduction linked terms. Genes with both H3K27me3 and H3K4me3 were annotated as differentiation and development associated terms. This was consistent with the findings that bivalent genes, which were enriched with both H3K27me3 and H3K4me3, were mainly differentiation marker genes in mESCs. We include this data at Page 8, last paragraph.

4. As shown in Fig. 2, the “Mix” cell population was identified as an intermediate between “Naïve” and “Prime” state. Mix cluster was enriched for differentiation -related term such as neuron interaction (Figure S8). Why differentiation terms were not observed in the “Prime”

population? Furthermore, Nanog and Rex1 broadly expressed in Mix cluster, which is not seen in Naïve cluster (Fig. S9). This is confusing.

In line with comment #5 the reviewer raised below, we thank the reviewer to point out this misleading interpretation. The pseudotime analysis showed a progression of Naïve cluster at the beginning of the trajectory to Prime cluster at the end. The Mix cluster separating from the pseudotime analysis was detected in one branch alongside the trajectory. This trajectory indicated a differentiation progress from the early Naïve cluster to late Prime cluster, whereas the Mix cluster occurred as a distinct differentiation lineage to differ from the asymmetrically differentiated cells. We modified the description of Mix cluster in the result section to avoid the misleading interpretation. Page 14, last paragraph. We also thank the reviewer for these comments about differentiation terms in GO analysis. We were promoted to look at the GO terms carefully and found that the GO terms in Prim cluster were annotated to several nucleoside triphosphate metabolic process terms. It's been reported that this process participated in erythroid differentiation², T cell lineage differentiation³, and neurogenesis⁴. So, the missing terms of differentiation might be overridden by the nucleoside triphosphate metabolic process terms. With the updated interpretations, as cells in Mix cluster were segregated into a distinct differentiation lineage, the Nanog and Rex1 levels in Mix cluster were low as compared to Naïve cells as shown in Fig. S9. We include these new data in Page 13, first paragraph.

5. In Fig. 2j, two trajectories were identified by monocle2. The authors should elaborate on the lineage relationship here, which is also an additional clarification of the “Mix” cells (mainly occur in one branch) as I mentioned above.

We thank the reviewer for this great advice. In line with the response above, we described more about the Mix cluster:

Proxi cluster was at the beginning of the trajectory and Dista cluster was at the end of the trajectory. In addition, the Mix cluster, which initiated in the middle of the differentiation stage, was detected in one branch alongside the trajectory. This trajectory indicated a differentiation progress from the early Proxi cluster to the late Dista cluster, whereas the Mix cluster occurred as a distinct differentiation lineage to differ from the asymmetrically divided cells. We also labeled cells with their relative positions to Wnt3a beads in the trajectories. Page 14, last paragraph.

6. Most of the Mix/Naïve/Prime cluster markers shown in Fig. 3a are not specific for each cluster both on the RNA and histone level.

We enlarged this panel to show histone mark signals and gene expression clearly. To improve the reading, we also added the significance labels for the RNA seq. The levels of histone marks were shown in enlarged view to improve the reading. We hope this new figure shows the specificity of histone marks, especially H3K27me3, and gene expression for each cluster.

7. Using module analysis to calculate the correlation between transcriptome and histone marks, the authors found that H3K27me3 was more correlated with gene expression. This is inconsistent with many previously reported studies, i.e. H3K4me3 is an active mark for gene expression. Given that very little is done to evaluate the single-cell data quality, I am rather concerned about the potential noise and data sparsity beyond the biological conclusions.

We thank the reviewer for this comment to improve data presentation. We agree that H3K4me3 is an active mark for gene expression and it is correlate with gene expression. By using the module analysis, we only stated that H3K4me3 was less correlated with gene expression than H3K27me3 in this specific case. And the ratio of significantly correlated modules was higher in H3K27me3 scSET-seq (21.5%) than H3K4me3 scSET-seq (10.7%).

In addition to the module analysis, we calculated the enrichment of histone marks at marker genes and clustered the cells using epigenomic profiles. These data indicate that H3K27me3 is highly correlated with the marker gene expression but not H3K4me3.

During early embryo development, the dynamic gain-and-loss of H3K4me3 and H3K27me3 happens at different embryogenesis stages. Cell context-dependent relationships among histone modifications are also important for the regulation of pluripotency and cell fate commitment, where different histone marks are dramatically changed at specific stages. When the Wnt signal is inhibited, mESCs are differentiated toward the epiblast stem cell-like (EpiLC) states. Previous reports have shown that, during the induction of mESCs to EpiLCs, histone modifications are largely reorganized¹⁰. EpiLCs exhibit abundant bivalent gene promoters with decreased H3K27me3 compared to mESCs. While EpiLCs are subsequently induced to differentiate into primordial germ cell-like cells (PGCLCs), H3K4me3 initially decreases at the differentiation genes but subsequently increases with concomitant elevated H3K27me3. Consistent with the transform from mESCs to EpiLCs, we also detect that, without WNT3a treatment in mESCs, H3K27me3 peaks are dramatically changed while H3K4me3 peaks are largely overlapped (Supplemental Fig. S7c). We include these in the discussion section, Page 25, first paragraph.

We hope, with the revised quality controls, information about the biological repeats and detailed method, the quality of the sequencing results is convincing.

Minor points:

1. The language demands sufficient work to improve it as the current format more or less prevents one from fully appreciating this work.

We referred to a native speaker and improved the language.

2. Many typos exist in the manuscript. For example, “stem cell-like statues” should be “stem cell-like status”; “Figure ledged” should be “Figure legend”; “virtualize” should be “visualize”.

We thank the reviewer for his/her time to read the manuscript and point out these typos. We do appreciate his/her time for this help. We corrected these typos and checked other parts of the manuscripts to improve the writing.

Reference:

1. Wang, X., He, Y., Zhang, Q., Ren, X. & Zhang, Z. Direct Comparative Analyses of 10X Genomics Chromium and Smart-seq2. *Genomics Proteomics Bioinformatics* (2021).
2. Chae, H.D., Lee, M.R. & Broxmeyer, H.E. 5-Aminoimidazole-4-carboxamide ribonucleoside induces G(1)/S arrest and Nanog downregulation via p53 and enhances erythroid differentiation. *Stem Cells* **30**, 140-149 (2012).
3. Almeida, L., Lochner, M., Berod, L. & Sparwasser, T. Metabolic pathways in T cell activation and lineage differentiation. *Semin Immunol* **28**, 514-524 (2016).
4. Langer, D., Ikehara, Y., Takebayashi, H., Hawkes, R. & Zimmermann, H. The ectonucleotidases alkaline phosphatase and nucleoside triphosphate diphosphohydrolase 2 are associated with subsets of progenitor cell populations in the mouse embryonic, postnatal and adult neurogenic zones. *Neuroscience* **150**, 863-879 (2007).

5. Conway, E. *et al.* A Family of Vertebrate-Specific Polycombs Encoded by the LCOR/LCORL Genes Balance PRC2 Subtype Activities. *Mol Cell* **70**, 408-421 e408 (2018).
6. Griizenhout, A. *et al.* Functional analysis of AEBP2, a PRC2 Polycomb protein, reveals a Trithorax phenotype in embryonic development and in ESCs. *Development* **143**, 2716-2723 (2016).
7. van Mierlo, G., Veenstra, G.J.C., Vermeulen, M. & Marks, H. The Complexity of PRC2 Subcomplexes. *Trends Cell Biol* **29**, 660-671 (2019).
8. Shen, X. *et al.* EZH1 mediates methylation on histone H3 lysine 27 and complements EZH2 in maintaining stem cell identity and executing pluripotency. *Mol Cell* **32**, 491-502 (2008).
9. Kasinath, V. *et al.* JARID2 and AEBP2 regulate PRC2 in the presence of H2AK119ub1 and other histone modifications. *Science* **371** (2021).
10. Kurimoto, K. *et al.* Quantitative Dynamics of Chromatin Remodeling during Germ Cell Specification from Mouse Embryonic Stem Cells. *Cell Stem Cell* **16**, 517-532 (2015).

Reviewers' Comments:

Reviewer #1:

Remarks to the Author:

The authors have addressed the majority of our comments, and the manuscript has been strengthened in several key areas. We have a small number of comments and queries that have come up in the revision and we feel should be addressed before publication:

1. In the new Supplementary Fig 6e that was introduced in the revised manuscript, it is surprising to see that the aggregated single cells detected far fewer peaks compared to those peaks identified in the aggregated single cell datasets (only ~10% H3K27me3, ~20% H3K4me3) and different peaks (~50% peaks defined in H3K27me3 single cell datasets are not found in either bulk SET or Encode datasets) compared to both bulk SET-seq and ENCODE. This causes challenges with interpreting the findings based on scSET-seq data. The authors should clarify and discuss this technical difference. As a comparison, aggregated Paired-Tag single cell datasets can find ~60% overlapped peaks of ENCODE datasets.

2. The labelling of 'Proxi' and 'Dista' for the clusters should be unified across the manuscript and figures. In quite a few places, particularly in the figures, the text is still labelled 'naive' and 'primed', such as Fig 2b-i, Fig 4c-f, Fig 5, and some supplemental figures.

3. The accuracy of several statements in the manuscript should be improved:

3.1 On page 6, first paragraph: 'The usage of Tn5 for tagmentation of cDNA', here we believe that 'cDNA' should be more accurately described as 'mRNA/cDNA hybrid'.

3.2 On page 9, first paragraph: 'This data indicates that, when bulk cells are used to detect both H3K4me3 and H3K27me3 along with gene expression signals, around 1,000 cells are optimal for this specific application'. But we don't see how the authors arrive at this conclusion – can you explain please?

3.3 On Page 10, second paragraph: 'This was simply because we sequenced more reads in the scSET-seq.' We agree this could be a major reason, but there could also be other reasons, and it would be helpful for the reader to understand these too.

Reviewer #2:

Remarks to the Author:

In general, the authors did a nice job in revising the manuscript. The newly added analyses, especially the comparison with published methods, are very helpful.

Here I just have two relatively minor points:

1. Regarding the Aebp2 knockout data, there is still a concern that the effects observed could be indirect. I wonder whether the rescue assay or acute knockout (not constitutively KO) is possible to avoid any potential secondary effects due to prolonged loss-of-function. At the very least, the authors could provide discussion on this possibility.

2. For the single cell collection, the authors mentioned that the dividing cells are traced with white light microscopy. Please provide recorded movies as a supplementary video, so that the readers would know exactly how these experiment was performed. This would be very helpful and avoid lengthy description.

Reviewer #3:

Remarks to the Author:

In this revision, Sun et al. have included some new analyses. Several clarifications and

comparisons with similar methods were added, strengthening the transparency in the SET-seq protocol. However, major concerns still remain.

1. As examples of IGV views of scSET-seq signals of histone marks in Fig 1g, bulk references as golden standard are lacking. In addition, the *Sim1* gene is not in the genomic region of 'chr10:50,699,363-50,780,024'. Further, I expect more representative genes in mESC to present here.

2. In Supplemental Table S3, the FRiP of Wnt scSET-seq data is ~ 0.15 , demonstrating low specificity of the data. What makes me more worried is that there are only 335 and 210 cells in H3K27me3 and H3K4me3 scSET-seq after quality control. Given this tremendous technical noise and data sparsity together with the intrinsic low throughput with this method, it is insufficient to reach the conclusion of stronger connection between H3K27me3 and asymmetric cell division events.

3. The authors claimed that "the trajectory indicated a differentiation progress from the early Naïve cluster to late Prime cluster, whereas the Mix cluster occurred as a distinct differentiation lineage to differ from the asymmetrically differentiated cells". However, the authors still regarded "Mix" population as an intermediate between naïve and primed cells for the pseudotime analyses (Fig. 2j, Fig. S11). Since these three clusters are very close and not well-separated in the UMAP, more evidence and clarification about the lineage relationship would be helpful.

4. Although the authors have toned down the conclusions about the causative role of *Aebp2* in the cell fate decision of Wnt-induced asymmetric cell division in this revision, no dramatic changes about the ratio of daughter cells asymmetrically expressing *Nanog* were observed in Fig. 6d. Also, if the Mix cluster represents a different lineage compared with typical asymmetrically divided trajectory, what's the specific role of *Aebp2* in the Mix population?

5. Based on the limited evidence in Fig. 6, the title "Joint single-cell multiomic analysis identifies H3K27me3 as a key regulator in Wnt3a induced asymmetric stem cell division" should be accordingly modified. H3K27me3 as a key regulator can be barely supported given the marginal change identified here.

Due to these concerns about data quality and biological insights, I continue to think major conclusions are not well supported in this revision.

We thank the reviewers for their time and efforts to help us improve this manuscript. We performed additional analysis to address the comments and provided additional information to strength our conclusions. Please see the detailed response to every point below.

Reviewer #1 (Remarks to the Author):

The authors have addressed the majority of our comments, and the manuscript has been strengthened in several key areas. We have a small number of comments and queries that have come up in the revision and we feel should be addressed before publication:

1. In the new Supplementary Fig 6e that was introduced in the revised manuscript, it is surprising to see that the aggregated single cells detected far fewer peaks compared to those peaks identified in the aggregated single cell datasets (only ~10% H3K27me3, ~20% H3K4me3) and different peaks (~50% peaks defined in H3K27me3 single cell datasets are not found in either bulk SET or Encode datasets) compared to both bulk SET-seq and ENCODE. This causes challenges with interpreting the findings based on scSET-seq data. The authors should clarify and discuss this technical difference. As a comparison, aggregated Paired-Tag single cell datasets can find ~60% overlapped peaks of ENCODE datasets.

We thank the reviewer for this comment and help to improve the manuscript. We called peaks with a very stringent condition as $p < 0.01$, which eliminated most low-quality peaks. Under this condition, around 60% of peaks in scSET-seq overlapped with ENCODE peaks and bulk sequencing peaks. When we called peaks with a p value less than 0.05, the total number of peaks increased a lot in scSET-seq but not many in ENCODE and bulk SET-seq. The total number of peaks were similar among these datasets. Moreover, the ratios of overlapped scSET-seq peaks among different datasets were not dramatically changed, suggesting that lots of low enriched true peaks were eliminated when we used the high stringent condition (Letter Figure 1). In addition, ENCODE datasets in HeLa cells showed much more peaks (around 10-fold more) than those in mESCs. These high levels of histone modifications may also cause the different affinity of antibodies and subsequent peaks recapitulated. We kept the original high stringent condition to get a more solid conclusion. However, as the reviewer pointed out, a smaller number of peaks were called in scSET-seq when compared with ENCODE data. We keep the high stringent condition to call the true peaks and discussed this in the discussion section on page 23, second paragraph.

Letter Figure 1. Venn diagrams showing the overlaps of H3K27me3 and H3K4me3 SET-seq peaks from 10,000 cells, ENCODE data peaks, and merged scSET-seq peaks, respectively. Peaks were called with a soft quality $p < 0.05$. Fisher's exact statistical tests were done for the overlaps and p values were less than 0.01 between every two overlaps.

2. The labelling of 'Proxi' and 'Dista' for the clusters should be unified across the manuscript and figures. In quite a few places, particularly in the figures, the text is still labelled 'naive' and 'primed', such as Fig 2b-i, Fig 4c-f, Fig 5, and some supplemental figures.

We thank the reviewer to carefully check the text and figures. We corrected these labels in all the main figures and supplemental figures to get a unified description of the clusters. These figures, including Fig 2, Fig 4, Fig 5, Fig. S7, Fig. S8, Fig S10, Fig S11, and Fig S13, were corrected. The figure legend of supplemental figures were corrected accordingly.

3. The accuracy of several statements in the manuscript should be improved:

3.1 On page 6, first paragraph: 'The usage of Tn5 for tagmentation of cDNA', here we believe that 'cDNA' should be more accurately described as 'mRNA/cDNA hybrid'.

We thank the reviewer for this suggestion. We improved the accuracy of statements about mRNA/cDNA hybrid in this place. Other descriptions in the manuscript were also corrected to mRNA/cDNA hybrid.

3.2 On page 9, first paragraph: 'This data indicates that, when bulk cells are used to detect both H3K4me3 and H3K27me3 along with gene expression signals, around 1,000 cells are

optimal for this specific application'. But we don't see how the authors arrive at this conclusion – can you explain please?

We thank the reviewer to suggest us specify this. The sentence is misleading here. We corrected the sentence to "...at least around 1,000 cells are optimal...". We detected around 2,000 to 5,000 genes in one cluster when 1,000 or 10,000 cells were used. When 100 cells were used as the starting material, the genes with detected H3K27me3, H3K4me3 and gene expression signals dropped to around 300 to 800 in each cluster. So, we speculated that no less than 1,000 cells were optimal for this application. The numbers of genes in each cluster were labeled in the figure. And these data were included in the revised manuscript on page 9, first paragraph.

3.3 On Page 10, second paragraph: 'This was simply because we sequenced more reads in the scSET-seq.' We agree this could be a major reason, but there could also be other reasons, and it would be helpful for the reader to understand these too.

We thank the reviewer for this helpful suggestion. We sequenced more than 2 million reads per cell to reach the saturation of unique fragments recovered, while not many reads were sequenced per cell in the Paired-tag to get maximum recoveries of epigenomic sequencing results. The other possibility might be a slightly loss of signals when cells were barcoded by ligation of index primers, which were absent in the scSET-seq. These new explanations were included on page 11, first paragraph.

Reviewer #2 (Remarks to the Author):

In general, the authors did a nice job in revising the manuscript. The newly added analyses, especially the comparison with published methods, are very helpful.

Here I just have two relatively minor points:

1. Regarding the Aebp2 knockout data, there is still a concern that the effects observed could be indirect. I wonder whether the rescue assay or acute knockout (not constitutively KO) is possible to avoid any potential secondary effects due to prolonged loss-of-function. At the very least, the authors could provide discussion on this possibility.

We agree with the reviewer that the effects may not be a direct cause of Aebp2. We planned to do a rescue or acute knockout assay before the original submission. In the rescue assay, it

is critical to express Aebp2 at the original level which is important to maintain the cell fate determination. The acute knockout timing may affect the knockout efficiency. If it's too early to knockout Aebp2, the prolonged loss of Aebp2 may lead to second effect. If it's too late to knockout Aebp2 in single cells, the Aebp2 protein is not fully degraded. We sincerely agree that these assays, if performed perfectly, could provide new clues to dissect the effect of Aebp2 in the cell fate determination. We discussed these point in the discussion and provide this possibility. on page 28, first paragraph.

2. For the single cell collection, the authors mentioned that the dividing cells are traced with white light microscopy. Please provide recorded movies as a supplementary video, so that the readers would know exactly how these experiment was performed. This would be very helpful and avoid lengthy description.

We thank the reviewer for suggesting us upload a video. We followed the reviewer's suggestion and provided a representative video (supplemental movie S1) for the dividing cells that were traced with white light microscopy.

Reviewer #3 (Remarks to the Author):

In this revision, Sun et al. have included some new analyses. Several clarifications and comparisons with similar methods were added, strengthening the transparency in the SET-seq protocol. However, major concerns still remain.

1. As examples of IGV views of scSET-seq signals of histone marks in Fig 1g, bulk references as golden standard are lacking. In addition, the Sim1 gene is not in the genomic region of 'chr10:50,699,363-50,780,024'. Further, I expect more representative genes in mESC to present here.

We thank the reviewer to help us improve the data presenting. We include the ENCODE data in the IGV views as a bulk reference. The view around Sim1 gene is at the upstream of Sim1 gene to show an intergenic region. As the reviewer suggested, we updated the figure with other IGV views to show more regions.

2. In Supplemental Table S3, the FRiP of Wnt scSET-seq data is ~0.15, demonstrating low specificity of the data. What makes me more worried is that there are only 335 and 210 cells in H3K27me3 and H3K4me3 scSET-seq after quality control. Given this tremendous technical noise and data sparsity together with the intrinsic low throughput with this method,

it is insufficient to reach the conclusion of stronger connection between H3K27me3 and asymmetric cell division events.

We thank the reviewer for this helpful suggestion. The FRiPs were calculated without distinguishing proximal and distal cells whereas the peaks called for FRiP calculation were mixed with proximal and distal cells. The proximal and distal cells were treated with or without Wnt3a beads, so that the H3K27me3 and H3K4me3 enriched regions were different in these two groups of cells. When we calculated the FRiPs without dividing these two groups of cells, the peaks called didn't faithfully represent the real peaks. Like the method in Paired-tag which divided the cells by their original tissues, we recalculated the FRiPs by separating proximal and distal cells. The average FRiPs for H3K27me3 and H3K4me3 were around 0.33 and 0.34 respectively, which was similar to the Paired-tag. In addition, we used cisTopic to remove the potential noises before we incorporated the epigenomic signals into single cells. The calculated signals were more specific in the single cells. To further enrich the signals in one cell, we sequenced large amount of reads to reach a maximal recovery of unique fragments. The average detected unique fragments per cell were 70 to 200 times more than the Paired-Tag (Supplemental Fig. S6d) and other similar method, including CoBATCH, itChIP, and HT-scChIP-seq. Together, even though the cell numbers we detected were not so big as other similar method which were not perform in a specific biology system, the total signals were recovered to the similar levels.

The throughput of this method was similar to Smart-seq2 which can be used to profile several hounds of cells in one experiment. The number of detected cells was limited mainly by the asymmetric dividing system, in which the current technology could not determine the relative positions of cells to beads.

Moreover, the FRiPs were slightly higher for H3K4me3 than those for H3K27me3 in both scSET-seq and Paired-tag (Fig. 1h). If considering the noise of sequencing results, the accuracies of H3K27me3 would be lower than H3K4me3 and the decrease of correlations of H3K27m3 with gene expression would be larger than that of H3K4me3 with gene expression. Under this condition, we still found that H3K27me3 was highly correlated with gene expression changes in asymmetric cell division.

Through bulk sequencing data, we also detected that, with or without WNT3a treatment, H3K27me3 peaks were dramatically changed while H3K4me3 peaks were not (Supplemental Fig. S7c), suggesting a fast turnover of H3K27me3 with the WNT3a treatment.

We specified this separation of cells for FRiP calculation in the head of Supplemental Table S3. And we discussed this in the discussion section on page 23, last paragraph.

3. The authors claimed that “the trajectory indicated a differentiation progress from the early Naïve cluster to late Prime cluster, whereas the Mix cluster occurred as a distinct differentiation lineage to differ from the asymmetrically differentiated cells”. However, the authors still regarded “Mix” population as an intermediate between naïve and primed cells for the pseudotime analyses (Fig. 2j, Fig. S11). Since these three clusters are very close and not well-separated in the UMAP, more evidence and clarification about the lineage relationship would be helpful.

We thank the reviewer for the help to suggest us clarify this. The circles in figure 2j were used to indicate the clustered cells. We modified the figure to remove the circles to avoid this potential misleading impression. In addition, the clusters in original Fig. S11 were presented in one-dimensional way to show the changes of genes among these three clusters. The Mix cluster, which branched from Naïve cluster, was not an intermediate between Naïve and Prime clusters. We emphasized the relationships among clusters in the results section to avoid the misleading understandings of the figure. Page 15, first paragraph.

As the reviewer suggested, we also performed STREAM (Single-cell Trajectories Reconstruction, Exploration And Mapping) analysis¹ to assign single cells for the reconstruction of developmental trajectories. This analysis recapitulated the branched Mix cluster and early Naïve cluster to late Prime cluster, further supporting the accuracy of pseudotime reconstruction, further supporting the accuracy of pseudotime reconstruction (Supplemental Fig. S11a). This new result was included on page 15, first paragraph.

4. Although the authors have toned down the conclusions about the causative role of Aebp2 in the cell fate decision of Wnt-induced asymmetric cell division in this revision, no dramatic changes about the ratio of daughter cells asymmetrically expressing Nanog were observed in Fig. 6d. Also, if the Mix cluster represents a different lineage compared with typical asymmetrically divided trajectory, what’s the specific role of Aebp2 in the Mix population?

In Aebp2 KO cells, H3K27me3 increased at the peaks overlapped with Aebp2 peaks, which was consistent with previous results. This compensation of increased H3K27me3 may cause the observation that, when Aebp2 was knocked out, not all daughter cells asymmetrically expressed Nanog. Aebp2 was reported to be a regulator in PRC2 complex to direct the localization of PRC2 on chromatin and to regulate the enzymatic activities of PRC2. Aebp2 ChIP-seq peaks were detected at the promoters of 200/344 marker genes and it’s highly expressed in the Mix cluster, which exhibited a high enrichment of H3K27me3 at marker

genes. In Mix cluster, Aebp2 could increase H3K27me3 at marker genes to repress the activation of genes that were important for the asymmetrical cell division. We added these in the discussion section on page 28, first paragraph.

5. Based on the limited evidence in Fig. 6, the title “Joint single-cell multiomic analysis identifies H3K27me3 as a key regulator in Wnt3a induced asymmetric stem cell division” should be accordingly modified. H3K27me3 as a key regulator can be barely supported given the marginal change identified here.

We thank the reviewer for this suggestion to modify the title. We changed the title to “Joint single-cell multiomic analysis in Wnt3a induced asymmetric stem cell division”

Due to these concerns about data quality and biological insights, I continue to think major conclusions are not well supported in this revision.

We hope, with the new data, specifications and corrections, the concerns of reviewers are addressed.

Reference:

1. Chen, H. *et al.* Single-cell trajectories reconstruction, exploration and mapping of omics data with STREAM. *Nat Commun* **10**, 1903 (2019).

Reviewers' Comments:

Reviewer #1:

Remarks to the Author:

The authors have addressed my comments.

Reviewer #2:

Remarks to the Author:

I am OK with the revision and recommend acceptance of this work.

Reviewer #3:

Remarks to the Author:

In this revision, the authors provided more details in this method as well as several clarifications. I was astonished by the superb quality of single cells by scSET-seq because it basically shared the principle in the molecular design with other similar methods. Therefore, I was motivated to reanalyze the raw data as currently presented in Figure S6d provided by the authors. Surprisingly, my analysis identified the unique fragments per cell ~ 100 folds less than that presented in Figure S6d.

Overall, I was not convinced by the statement that "The average detected unique fragments per cell were 70 to 200 times more than the Paired-Tag (Supplemental Fig. S6d) and other similar method, including CoBATCH, itChIP, and HT-scChIP-seq". Importantly, I think the analysis as presented in Fig S6d (the comparison of the scSET-seq and Paired-Tag methods) is not comprehensive and not fair. The authors should do a good job by performing a fair comparison by analyzing all data in the same computational pipeline (for example their own pipeline).

In addition, the raw data provided by the authors in the dropbox link were from the single modal omics assay, yielding chromatin only, rather than dual omics (Chromatin + RNA) generated by the scSET-seq method. Nevertheless, this analysis already calls attention to the misinterpretation of the current data in this manuscript. I would suggest that the authors faithfully report the quality and results.

One key limitation of current single-cell multimodal omics methods is the data sparsity and potential noise. I do not think that the poor quality of chromatin part would be a major barrier to publication, but does need to be clearly presented on and fairly compared.

Reviewer #1 (Remarks to the Author):

The authors have addressed my comments.

We would like to thank reviewers' insightful comments and time for reviewing this exciting story.

Reviewer #2 (Remarks to the Author):

I am OK with the revision and recommend acceptance of this work.

We thank reviewer for reviewing this manuscript and encouraging comments.

Reviewer #3 (Remarks to the Author):

In this revision, the authors provided more details in this method as well as several clarifications. I was astonished by the superb quality of single cells by scSET-seq because it basically shared the principle in the molecular design with other similar methods. Therefore, I was motivated to reanalyze the raw data as currently presented in Figure S6d provided by the authors. Surprisingly, my analysis identified the unique fragments per cell ~100 folds less than that presented in Figure S6d.

Overall, I was not convinced by the statement that "The average detected unique fragments per cell were 70 to 200 times more than the Paired-Tag (Supplemental Fig. S6d) and other similar method, including CoBATCH, itChIP, and HT-scChIP-seq". Importantly, I think the analysis as presented in Fig S6d (the comparison of the scSET-seq and Paired-Tag methods) is not comprehensive and not fair. The authors should do a good job by performing a fair comparison by analyzing all data in the same computational pipeline (for example their own pipeline).

In addition, the raw data provided by the authors in the dropbox link were from the single modal omics assay, yielding chromatin only, rather than dual omics (Chromatin + RNA) generated by the scSET-seq method. Nevertheless, this analysis already calls attention to the misinterpretation of the current data in this manuscript. I would suggest that the authors faithfully report the quality and results.

One key limitation of current single-cell multimodal omics methods is the data sparsity and potential noise. I do not think that the poor quality of chromatin part would be a major barrier to publication, but does need to be clearly presented on and fairly compared.

We thank the reviewer for his/her time and efforts to help us improve the manuscript. These comments helped us clarify the statements.

We agree that the chromatin sequencing method shared the same principle in the molecular design. We also agree that the total unique fragments per cells were expected to be similar between scSET-seq and Paired-tag. We tried to emphasize that this difference is

not because of sequencing method but because of the depth of library sequenced in the results section. We never argued that this method was better than others, and we tried to explain it clearly to avoid the misunderstanding. When we presented this data in the results section, we immediately specified that this difference was likely just because more reads were sequenced. The results section: Page 11, first paragraph : “We sequenced more than 2 million reads per cell to reach the saturation of unique fragments recovered, while not many reads were sequenced per cell in the Paired-tag to get maximum recoveries of epigenomic sequencing results. This increase of unique fragments may simply because we sequenced more reads in the scSET-seq. The other possibility might be a slightly loss of signals when cells were barcoded by ligation of index primers, which were absent in the scSET-seq.”

There are two major difference between the reviewer’s calculation and ours: 1. The total reads in each cell. 2. The ratio of unique fragments compared to mapped reads. Beyond that, we calculated similar mapping ratios. We explained how we performed this analysis and put the code below:

For the point 1, the total reads in each cell.

We demultiplexed the sequencing results by first cutting off the common sequence before cell index, and then using fastq-multx to separate the reads into each cell through cell index. These reads were further polished by removing the common transposome sequences and then subjected to quality control and reads mapping. Code was attached at the end.

For the point 2, The ratio of unique fragments compared to mapped reads.

In scSET-seq pipeline, we used Picard to remove duplicated reads. The unique fragments in Supplemental Fig. S6d were calculated by using Paired-tag pipeline where reachtools was used (Please note, the determine statements of UMI were removed in the code of reachtools to avoid false detection of unique fragments). We communicated with Dr. Ren and got the pipeline they used to calculate the unique fragments. During the revision, we then used Paired-tag codes to do the analysis for a comparison. The reads mapping and unique fragment calculation were performed with the same pipeline.

Originally, as the reviewer suggested, we used the same code of data presenting in scSET-seq and Paired-tag. In scSET-seq, both reads were target reads. In Paired-tag, one read contained target reads and the other read represented the index information. If Picard was used as the program for duplication removal in Paired-tag, the UMI was not token into count, leading to false removal of unique fragments. The calculated unique fragments would be less than the true unique fragments. Together, scSET-seq pipeline couldn’t fit Paired-tag, but Paired-tag could be used in scSET-seq. So, we didn’t use our pipeline to re-calculate Paired-tag data, but used Paired-tag pipeline to calculate the unique fragments in scSET-seq.

However, we re-calculated the unique fragments by scSET-seq pipeline, where Picard was used. We found that, if using Picard in scSET-seq analysis, the unique fragments were much less than the ones defined by reachtools which was around 100 times less (Letter Fig. 1). Moreover, the numbers of unique fragments were similar between scSET-seq and Paired-Tag.

scSET-seq						
H3K27me3	total reads	mapping	unique_read	Picard_unique_fragments	reachtools_unqie_fragments	
top 120 cells	9941460	0.04	291486	4773	219791	
top 240 cells	7156363	0.04	213292	3528	160596	
480 cells	4473061	0.05	143282	2390	106946	
H3K4me3	total reads	mapping	unique_read	Picard_unique_fragments	reachtools_unqie_fragments	
top 120 cells	13138300	0.11	887010	9890	651476	
top 240 cells	8568856	0.14	715255	8747	527026	
480 cells	5270436	0.13	467192	6449	344494	
Paired-Tag						
H3K27me3	total reads	mapping	unique_read	Picard_unique_fragments	reachtools_unqie_fragments	by Pair-Tag author
top 120 cells	27818	0.84	20582	5110	6995	6863
top 240 cells	20906	0.81	15105	3726	5154	5082
480 cells	15083	0.77	10531	2591	3623	3570
H3K4me3	total reads	mapping	unique_read	Picard_unique_fragments	reachtools_unqie_fragments	by Pair-Tag author
top 120 cells	93218	0.83	70090	9275	13398	13425
top 240 cells	69587	0.83	52586	6893	10011	10030
480 cells	50943	0.83	38468	5008	7308	7324

Letter Figure 1. Sequencing reads quality analysis in scSET-seq.

Properly, a more fitting computer program, but not the same pipeline, would be better and fairer for this comparison. So, we re-draw Supplemental Fig. S6 with different duplication removing method: Picard for scSET-seq and reachtools for Paired-Tag. In addition, to increase the strength of comparison, cells with top 400 total sequencing reads were plotted in Supplemental Fig. S6d.

The Dropbox link contains all the sequencing results including Chromatin + RNA. In addition, the average unique fragments were 11613 and 4393 in H3K27me3 scSET-seq and H3K4me3 scSET-seq in Wnt induced cell division, respectively. The numbers of detected unique fragments were similar to the chromatin data alone.

We updated the numbers of unique fragments in Supplemental Fig. S6, Table S1, and Table S3 with the re-calculated data using scSET-seq pipeline for counting of unique fragments. The difference of how duplicate reads were detected were marked in Supplemental Fig. S6. All other data were calculated based on the Picard duplication removal data, so other data were not affected. Accordingly, we modified the sentence in Results section and Discussion section in Page 10, last paragraph and Page 23, last paragraph. Acknowledge section was changed accordingly, since we requested coding pipeline from Dr. Ren.

We hope, with these new data and explanations, we addressed the reviewer's concerns.

Code for splitting reads into single cells:

```
##### split fastq by barcode
### trim primer
for dir in *Epi*
do
cd $dir
mkdir result
for data in *_2.fq.gz
do
    data1=${data%%_2.fq.gz}_1.fq.gz
    newdata=${data%%.fq.gz}.fq
    newdata1=${data1%%.fq.gz}.fq
    trim_galore -j 24 --dont_gzip --hardtrim3 136 $data1
    trim_galore -j 24 --dont_gzip --hardtrim3 135 $data
    mv ${data1%%.fq.gz}.136bp_3prime.fq $newdata1
    mv ${data%%.fq.gz}.135bp_3prime.fq $newdata
done
### use fastq-multx to do splitting
fastq-multx -B ${bcpath}/wnt-96-chip.txt -m 2 -x -b *_1.fq *_2.fq -o ./result/%_1.fq -
o ./result/%_2.fq
rm -r *.fq
#### trim barcode and following primer
cd result
    for data in *_2.fq
    do
        data1=${data%%_2.fq}_1.fq
        newdata=$data
        newdata1=$data1
        trim_galore -j 24 --dont_gzip --hardtrim3 94 $data1
        trim_galore -j 24 --dont_gzip --hardtrim3 93 $data
        rm $data1 $data
        mv ${data1%%.fq}.94bp_3prime.fq $newdata1
        mv ${data%%.fq}.93bp_3prime.fq $newdata
    done
cd ..
cd ..
done
#####
```

Reviewers' Comments:

Reviewer #3:

Remarks to the Author:

I pretty much appreciate the authors' interpretation for the discrepancy between their analysis and mine. That being said, "We sequenced more than 2 million reads per cell to reach the saturation of unique fragments recovered, while not many reads were sequenced per cell in the Paired-tag to get maximum recoveries of epigenomic sequencing results. This increase of unique fragments may simply because we sequenced more reads in the scSET-seq. The other possibility might be a slightly loss of signals when cells were barcoded by ligation of index primers, which were absent in the scSET-seq." However, I don't think that sequencing up to 2 million reads per cell can increase the unique fragments by ~ 100 folds since most cells are already saturated at the given sequencing depth of $\sim 200,000$ reads per cell with the duplicate rate of $\sim 99\%$. In addition, scSET-seq data present extremely low mapping rate ($\sim 0.04-0.05\%$), suggesting a vast majority of reads are useless. Thus, I strongly suggest that this type of speculation be excluded in the text.

Further, the authors need include the average and median fragments per cell as reported in Fig. S6 in Page 10 in the main text.

Reviewer #3 (Remarks to the Author):

I pretty much appreciate the authors' interpretation for the discrepancy between their analysis and mine. That being said, "We sequenced more than 2 million reads per cell to reach the saturation of unique fragments recovered, while not many reads were sequenced per cell in the Paired-tag to get maximum recoveries of epigenomic sequencing results. This increase of unique fragments may simply because we sequenced more reads in the scSET-seq. The other possibility might be a slightly loss of signals when cells were barcoded by ligation of index primers, which were absent in the scSET-seq." However, I don't think that sequencing up to 2 million reads per cell can increase the unique fragments by ~100 folds since most cells are already saturated at the given sequencing depth of ~200,000 reads per cell with the duplicate rate of ~99%. In addition, scSET-seq data present extremely low mapping rate (~0.04-0.05%), suggesting a vast majority of reads are useless. Thus, I strongly suggest that this type of speculation be excluded in the text. Further, the authors need include the average and median fragments per cell as reported in Fig. S6 in Page 10 in the main text.

We thank the reviewer for this advice and we agree with the reviewer's comments that this speculation should be removed. We have already excluded this speculation about the difference from the results section in our last revision. In addition, we included the average and median fragments per cell in Supplemental Figure S6d in the main text. Page 10, last paragraph.